# Lack of bombesin receptor–activated protein attenuates bleomycin-induced pulmonary fibrosis in mice

Hui Wang[1], Wenrui Zhang[1], Rujiao Liu[1], Jiaoyun Zheng[2], Xueping Yao[1], Hui Chen[1], Jie Wang[1], Horst Christian Weber[3] , Xiaoqun Qin[1], Yang Xiang[1], Chi Liu[1], Huijun Liu[1], Lang Pan[1], Xiangping Qu[1]

**Bombesin receptor–activated protein (BRAP) was found to express in the interstitial cells of human fibrotic lungs with unknown function. Its homologous protein, encoded by *BC004004* gene, was also present in mouse lung tissues. We used *BC004004*[−/−] mice which lack BRAP homologous protein expression to establish a bleomycin-induced lung fibrotic model. After bleomycin treatment, *BC004004*[−/−] mice exhibited attenuation of pulmonary injury and less pulmonary fibrosis. Fibroblasts from *BC004004*[−/−] mice proliferated at a lower rate and produced less collagen. Autophagy-related gene 5 (ATG5) was identified as a partner interacting with human BRAP. Lacking BRAP homologous protein led to enhanced autophagy activity in mouse lung tissues as well as in isolated lung fibroblasts, indicating a negative regulatory role of this protein in autophagy via interaction with ATG5. Enhanced autophagy process in fibroblasts due to lack of BRAP homologous protein might contribute to the resistance of *BC004004*[−/−] mice to pulmonary fibrosis.**

## Introduction

Fibrosis can either be a primary progressive disease or present in many chronic inflammatory conditions that may ultimately lead to organ dysfunction. Fibrosis is featured by excessive deposition of fibrotic ECM proteins, especially type I collagen, which is produced mainly by accumulated activated fibroblasts in some organs such as lungs (Wynn, 2011; Mouw et al, 2014; White, 2015; Burgstaller et al, 2017). Despite intensive investigation regarding the underlying mechanisms of fibrosis, the pathogenesis of tissue fibrosis is still not fully understood. It was proposed that pulmonary fibrosis sometimes occurs via inflammatory and fibroproliferative response to lung injury (Hinz et al, 2007; Kendall & Feghali-Bostwick, 2014). Transforming growth factor-β1 (TGF-β1), released by airway epithelial cells and other cells in response to lung tissue injury, promotes the activation of fibroblasts and their subsequent differentiation into myofibroblasts and thus results in wound healing process (Gurtner et al, 2008; Wynn & Ramalingam, 2012; Palumbo-Zerr et al, 2015). But the responses to injuries can turn into pathological changes which mark the onset of fibrogenic process that finally develops pulmonary fibrosis. A more thorough understanding of the fibrogenic process requires the investigation of novel regulatory events at the cellular and molecular level.

The data from our recent study using a bleomycin-induced fibrotic lung injury mouse model demonstrate that the homologous protein of bombesin receptor–activated protein (BRAP) might be a negative regulator of pulmonary fibrosis. Human BRAP protein was first identified as a molecule that is a potential partner of the bombesin receptor subtype-3 (BRS-3) in a bacterial two-hybrid screen (Liu et al, 2011). It is encoded by human *C6orf89* gene and contains 354 amino acids. It was predicated to be a type II membrane protein with a putative N-terminal transmembrane (TM) domain (Lalioti et al, 2013) and a potential catalytic region within its C terminus (Liu et al, 2016). It is expressed in several cell types including bronchial epithelial cells, macrophages and neurons with functions that are still poorly characterized (data not shown). The overexpression of this protein in a cultured immortalized human bronchial epithelial cell line 16HBE14o- was found to promote proliferation of the cells and be involved in regulating transcriptional activity of NF-κB in those cells (Liu et al, 2016). In an attempt to further investigate the biological role of BRAP, we constructed a gene knockout mouse *BC004004*[−/−] that lacks the expression of the homologous protein of BRAP which is encoded by the mouse gene *BC004004*. This homologous protein of BRAP shares 83% identity with human BRAP. There are no significant histological changes of important organs including heart, lung, liver, kidney and spleen in *BC004004*[−/−] mice compared with the wild-type control mice as revealed by histological analyses. However, when manipulating the tissues of those mice we found that the fibrous connective tissue surrounding the bones and the joints is not as tough as that of the wild-type mice, which indicates a possible defect in fibrillar collagens in ECM due to lack of BRAP homologous protein. In addition, human BRAP protein was present in many interstitial cells in fibrotic

---

[1]Department of Physiology, School of Basic Medical Science, Central South University, Changsha, China   [2]Department of Pathlogy, The Second Xiangya Hospital, Central South University, Changsha, China   [3]Department of Pathology and Laboratory Medicine, Boston University School of Medicine, Section of Gastroenterology, Boston, MA, USA

Correspondence: quxiangping@csu.edu.cn

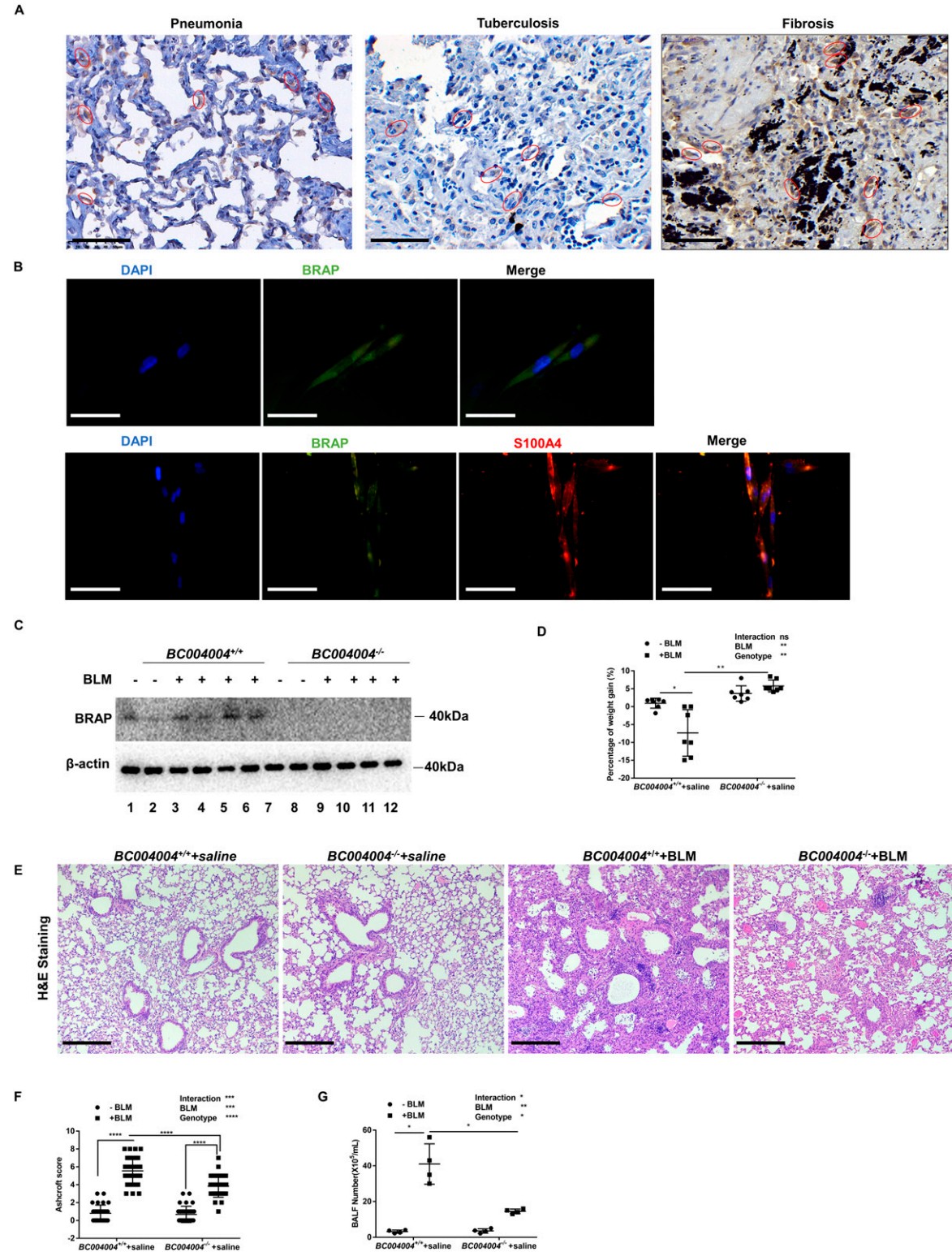

**Figure 1. Bombesin receptor–activated protein (BRAP) homologous protein deficiency attenuated bleomycin-induced pulmonary inflammation in mice.**
**(A)** Representative images of BRAP immunostaining in human tissue sections. Interstitial cells were indicated by red circles (original magnification ×400). Scale bar = 50 μm. **(B)** Representative images of immunofluorescence staining in cultured human lung fibroblast (HLF) cells. Immunostaining with anti-BRAP antibody (green color) was shown in the upper panel. Nuclei of the cells were localized by DAPI (blue). Double immunofluorescence staining in HLF cells was shown in the lower panel. Immunostaining using BRAP antibody was shown as green fluorescence and immunostaining using S100A4 antibody was shown as red fluorescence. The orange signals in the merged picture indicate the co-distribution of BRAP and S100A4 in the cytoplasm of fibroblasts (original magnification ×400). Bar = 50 μm. **(C)** BRAP homologous

lung tissues as shown by immunohistochemistry analyses with an anti-BRAP antibody. Therefore, we established a fibrotic lung injury model by treating $BC004004^{-/-}$ mice with bleomycin and found that the mice lacking BRAP homologous protein are more resistant to bleomycin-induced fibrotic lung injury. To further elucidate the underlying mechanism, we performed a yeast two-hybrid screen and identified ATG5, a key factor of autophagy, as an interacting partner of BRAP. We found that lacking BRAP homologous protein enhances autophagy in both isolated lung fibroblasts and in tissue sections of lungs from $BC004004^{-/-}$ mice. Together with other analyses our data suggest a link between autophagy and collagen production by fibroblasts. BRAP homologous protein could inhibit autophagy via interaction with ATG5. Enhanced autophagy process in fibroblasts due to lack of BRAP homologous protein might contribute to inhibition of pulmonary fibrotic injury in mice.

# Results

### BRAP and its homologous protein are expressed in lung interstitial cells

As shown in Fig 1A, BRAP was detected in lung interstitial cells on human tissue samples with pulmonary fibrosis as well as lung tissue with infection of pneumonia by immunohistochemistry staining using a rabbit monoclonal antibody against human BRAP (EPR13621 Cat. no. ab181073; Abcam). The staining with BRAP antibody in the interstitial cells on tissue samples with tuberculosis was not as significant as that of pneumonia and pulmonary fibrosis. We also found that BRAP was expressed in human lung fibroblast cell line HLF cells (Fig 1B, the upper panel). To examine the co-distribution of BRAP and fibroblast-specific protein 1 (FSP1, also called S100A4), which is considered a marker of fibroblasts, we performed double immunofluorescence analysis in cultured HLF cells. As shown in the lower panel of Fig 1B, green BRAP immunofluorescence overlapped with red S100A4 fluorescence and then the merged picture showed an orange signal which indicates the co-distribution of BRAP and S100A4 in the cytoplasm of fibroblasts. The presence of BRAP in interstitial cells of human fibrotic lung tissues indicates that this protein may have a regulatory role in pulmonary fibrosis. Western blotting analysis using the above anti-BRAP antibody also detected BRAP homologous protein in the lung tissues of wild-type mice due to the antibody's cross reactivity with mouse homologous protein (Fig 1C). Extracts from lung tissues of $BC004004^{-/-}$ mice (the strategy of genome engineering for creating $BC004004^{-/-}$ was illustrated in Fig S1) did not show any BRAP-positive staining by Western blotting.

### Lacking BRAP homologous protein in mice leads to attenuation of bleomycin-induced pulmonary injury in mice

To determine the mechanism of BRAP homologous protein during lung injury, we used a well-recognized model of lung fibrosis induced by the antineoplastic antibiotic bleomycin (Orlando et al, 2019). Intratracheal instillation of bleomycin led to pulmonary injury to the mice. On Day 21 after instillation, the body weights of wild-type control mice $BC004004^{+/+}$ dropped significantly (Fig 1D). However, $BC004004^{-/-}$ mice did not show significant body weight decrease on Day 21.

Bleomycin-induced injury to lung tissues was shown by H&E staining on Day 21 after intratracheal instillation as featured by collapse of alveolar, destruction of parenchyma and infiltration of cells in lung tissue sections. The infiltrations of inflammatory cells and destructions of parenchyma in lungs from $BC004004^{+/+}$ mice on Day 21 after bleomycin instillation were more severe compared with lung tissues of $BC004004^{-/-}$ mice (Fig 1E, the H&E staining images from isogenic controls which did not receive any treatment were provided in the Fig S2), which collaborates with the difference of body weight changes between knockout mice and wild-type control mice. Ashcroft scores were determined for all the 31 animals in each group to assess the severity of lung injury induced by bleomycin. As shown in Fig 1F, bleomycin induced more severe injury to lungs of wild-type mice compared with knockout mice by Ashcroft scoring analysis.

We performed bronchoalveolar lavage for four mice randomly selected from each group on Day 21 after bleomycin instillation, collected the bronchoalveolar lavage fluids (BALF) and counted the total numbers of the cells in BALF (Fig 1G). Bleomycin treatment led to significantly more cells in BALF of $BC004004^{+/+}$ mice compared with $BC004004^{-/-}$ mice, indicating an attenuation of the inflammation of lungs by BRAP homologous protein deficiency.

Bleomycin-induced fibrotic alterations of lung tissues, visible as accumulation of total collagen in the alveolar septa in both $BC004004^{+/+}$ and $BC004004^{-/-}$ mice, were revealed by Masson's trichrome and Sirius Red staining (Fig 2A and B). The obtained images were analyzed for area of positive staining which was measured and calculated as the percentage of total area for either Masson's trichrome staining or Sirius Red staining. Bleomycin-initiated pulmonary fibrosis was present in both $BC004004^{+/+}$ and $BC004004^{-/-}$ mice; however, accumulation of total collagen was much less severe in $BC004004^{-/-}$ mice as measured by both methods. Type I collagen is a fibrillar-type collagen in interstitial matrix and is by far the most abundant protein in all vertebrates. By immunohistochemistry (IHC) staining with an antibody against COL1A1, type I collagen was visualized on the tissue sections from bleomycin-treated mice (Fig 2C). The content of type I collagen

---

protein was present in lung tissues as revealed by Western blotting in wild-type control mice $BC004004^{+/+}$ with or without bleomycin treatment. The sample from an individual animal was loaded in one well. There was no BRAP homologous protein detected in lung tissues from $BC004004^{-/-}$ mice. The expression of β-actin was detected as a loading control. BLM, bleomycin. **(D)** The body weight changes were shown as percentages of weight gain on Day 21 after bleomycin instillation compared with the body weights on Day 0. Data are presented as mean ± SD; n = 7. *$P$ < 0.05, **$P$ < 0.01. **(E)** Representative histological changes of lung tissues after bleomycin treatment in both $BC004004^{-/-}$ mice and their wild-type control mice (original magnification ×200). Scale bar = 100 $\mu$m. **(F)** Ashcroft scores were determined for each tissue section (one tissue section from one mouse). Data are presented as mean ± SD; n = 31 for each group. ****$P$ < 0.0001. **(G)** The total cell numbers in bronchoalveolar lavage fluids (BALF) from mice on Day 21 after bleomycin instillation. There are more cells in BALF from $BC004004^{+/+}$ mice compared with $BC004004^{-/-}$ mice after bleomycin treatment (*$P$ = 0.0451, n = 4). Data are presented as mean ± SD. n = 4. *$P$ < 0.05, **$P$ < 0.01.
Source data are available for this figure.

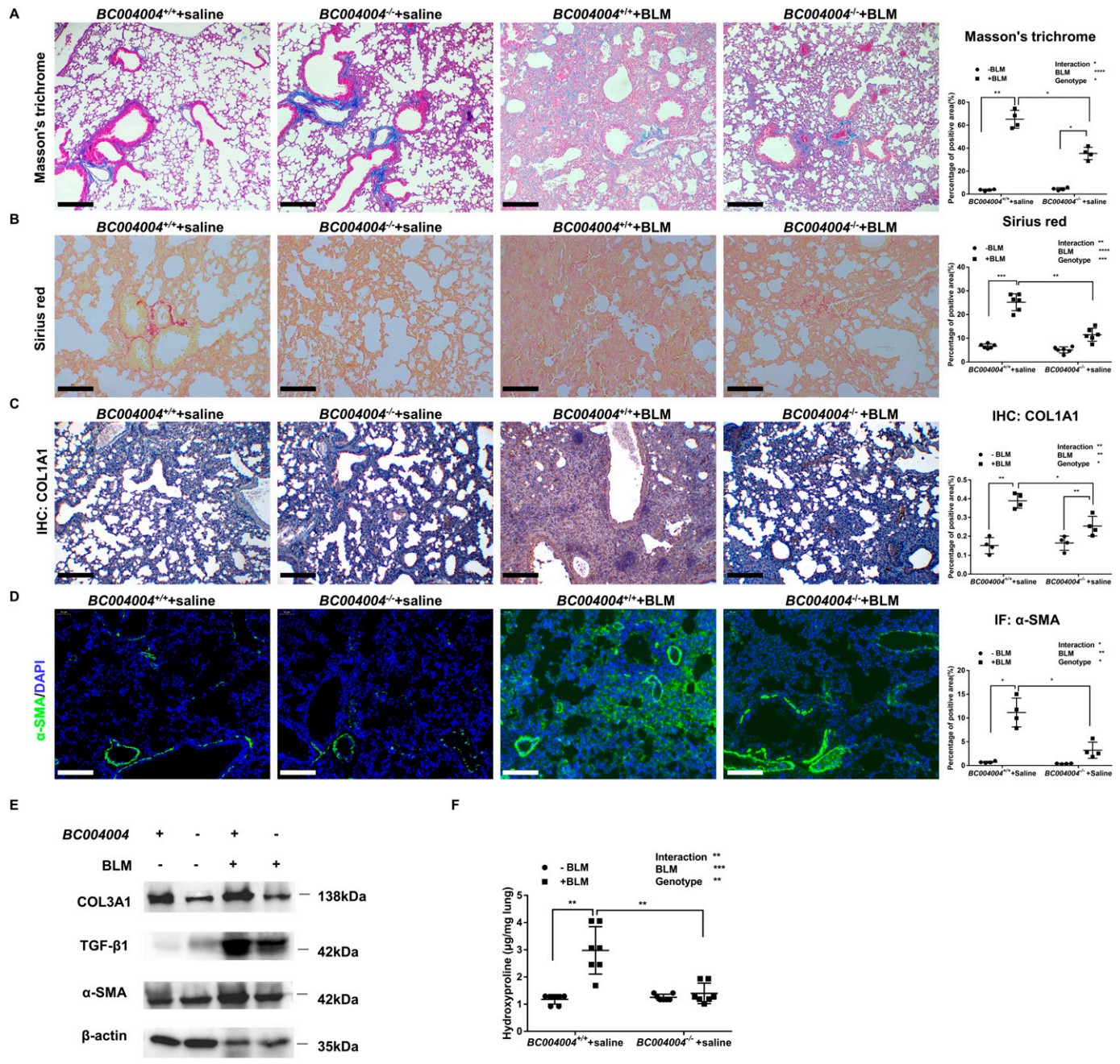

**Figure 2. Bleomycin-induced pulmonary fibrosis in mice was reduced in the absence of BC004004 expression.**
**(A)** Representative photomicrographs of Masson's trichrome stained lungs of mice on Day 21 after bleomycin administration (original magnification ×200). n = 4 in each group. Scale bar = 100 μm. **(B)** Representative photomicrographs of Sirius Red staining of lung tissues on Day 21 after bleomycin administration (original magnification ×200). n = 6 in each group. Scale bar = 100 μm. **(C)** Representative photomicrographs of COL1A1 immunostaining in mouse lung tissues (original magnification ×200). n = 4 in each group. Scale bar = 100 μm. **(D)** Immunofluorescence assay of mouse lung tissue sections on Day 21 after bleomycin instillation using an anti–α-smooth muscle actin primary antibody and a FITC fluorescent secondary antibody (original magnification ×200). n = 4 in each group. Scale bar = 100 μm. The areas of positive staining were measured and calculated as the percentage of total tissue areas (%) and presented on the right side to each panel (one tissue section from one mouse). **(E)** Representative images of Western blotting analysis of protein extracts of lung tissues on Day 21 after bleomycin instillation. n = 4 in each group. Bleomycin treatment led to less increase of TGF-β1 expression in lungs of BC004004$^{-/-}$ mice. **(F)** Hydroxyproline content in lungs of BC004004$^{+/+}$ mice and BC004004$^{-/-}$ mice with or without bleomycin administration. Data are presented as mean ± SD; n = 7 in each group. For (A, B, C, D, E, F), *P < 0.05, **P < 0.01, ***P < 0.001, ****P < 0.0001.
Source data are available for this figure.

protein increased significantly in lungs of BC004004$^{+/+}$ after bleomycin instillation. But the increase of type I collagen was much less in BC004004$^{-/-}$ mice after bleomycin instillation compared with

BC004004$^{+/+}$. Collectively, those results demonstrated that fibrotic lesions were much less severe after bleomycin treatment due to BRAP homologous protein deficiency.

Injuries to the airway epithelia may lead to the release of TGF-$\beta$1, a profibrotic cytokine that activates and transdifferentiates fibroblasts into myofibroblasts. The latter exhibits increased ECM secretion, in particular collagens. We therefore examined the levels of TGF-$\beta$1 and $\alpha$-smooth muscle actin ($\alpha$-SMA) which is expressed by myofibroblasts in the lungs of bleomycin treated mice. Western blotting showed significant increase of TGF-$\beta$1 expression induced by bleomycin in the protein extract of the lung tissues (Fig 2E). However, the expression level of TGF-$\beta$1 in $BC004004^{-/-}$ mice after bleomycin instillation was much less than that of $BC004004^{+/+}$ mice. Immunofluorescence assays with a primary anti-$\alpha$-SMA antibody revealed that the fluorescence of anti–$\alpha$-SMA was increased significantly in lungs of $BC004004^{+/+}$ after bleomycin instillation (Fig 2D). But the increase of anti–$\alpha$-SMA fluorescence was much less in $BC004004^{-/-}$ mice after bleomycin instillation compared with wild-type control mice. Western blotting with the protein extracts of the lungs showed that the increase of $\alpha$-SMA expression in $BC004004^{-/-}$ mice after bleomycin instillation is less than that of wild-type control mice, which collaborates with the results of immunofluorescence assay (Fig 2E). In addition, Western blotting using an antibody against COL3A1 showed that the content of type III collagen was also less in $BC004004^{-/-}$ mice even after bleomycin instillation compared with $BC004004^{+/+}$ (Fig 2E). Type III collagen is found in many tissues together with type I collagen. Changes in whole lung content of collagen, a major ECM component of fibrosis, was also evaluated by measuring hydroxyproline amount in the lung. As shown in Fig 2F, lung hydroxyproline content increased significantly in the wild-type control mice on day 21 after bleomycin administration, but bleomycin administration led to only a slight degree of increase of the lung hydroxyproline content in $BC004004^{-/-}$ mice. Taken together, the decreased levels of collagen I and III suggest attenuated fibrotic lesion in lungs of $BC004004^{-/-}$ mice after bleomycin instillation compared with wild-type controls. The expression of TGF-$\beta$1 of $BC004004^{-/-}$ mice cannot be increased as efficiently as the wild-type control after bleomycin treatment, which might be one of the reasons that account for the attenuated fibrotic lesions in those knockout mice.

### Primary fibroblasts isolated from lungs of $BC004004^{-/-}$ mice exhibited decreased collagen production and cell proliferation activity

In the development of pulmonary fibrosis, profibrotic factor TGF-$\beta$1 promotes the proliferation of fibroblasts and stimulates conversion of fibroblasts into myofibroblasts which produce a large amount of extracellular fibrous matrix. We isolated primary fibroblasts from lungs of $BC004004^{-/-}$ mice and found that the fibroblasts without BRAP homologous protein expression had lower cell proliferation activity at 48 and 72 h after TGF-$\beta$1 stimulation than the cells from wild-type control (Fig 3A). In accordance, flow cytometry analysis revealed a decrease in the percentage of $BC004004^{-/-}$ fibroblast cells in the S phase and an increase in the G1 phase as compared with fibroblasts from wild-type control mice (Fig 3B). The fibroblast cells were also stained with annexin V-FITC and propidium iodide (PI) and analyzed by flow cytometry to determine the number of cells undergoing apoptosis (Fig 3C). Most of the analyzed cells were viable and not undergoing apoptosis (lower left quadrant, both

annexin V-FITC and PI negative). The population of cells within upper left quadrant was observed as annexin V–FITC positive and PI negative and was considered as viable and undergoing early-phase apoptosis. The cells of upper right quadrant were considered to be dead cells or in the end stage of apoptosis (both annexin V-FITC and PI positive). There was an increase in the percentage of fibroblasts undergoing early-phase apoptosis from $BC004004^{-/-}$ mice. After TGF-$\beta$1 stimulation the percentage of $BC004004^{-/-}$cells undergoing early apoptosis were still higher than that of wild-type control cells. Taken together, the above analysis of isolated primary fibroblasts from lungs of mice indicates that the total number of fibroblasts that are able to produce collagen decreases in $BC004004^{-/-}$ mice.

Next, we tried to assess the ability of the isolated fibroblasts from knockout mice to produce collagens. The hydroxyproline content of the fibroblasts without BRAP homologous protein expression did not increase significantly in response to TGF-$\beta$1 stimulation, whereas the cells from wild-type control mice produced much more hydroxyproline content after TGF-$\beta$1 stimulation (Fig 3D). Western blotting shows that the fibroblasts from $BC004004^{-/-}$ mice contained less collagen $\alpha$-(I) chain (encoded by $COL1A2$ gene) than the cells from $BC004004^{+/+}$ mice (Fig 3E). The expression of $\alpha$-SMA in the fibroblasts from $BC004004^{-/-}$ mice also decreased compared with the wild-type control (Fig 3E). The TGF-$\beta$1 treatment of isolated fibroblasts induced transcriptional up-regulation of TGF-$\beta$1 in those cells because the mRNA levels were increased remarkably in both wild-type control mice and knockout mice as shown in Fig 3F. After being stimulated by exogenous TGF-$\beta$1, there is no significant difference for the mRNA expression level of TGF-$\beta$1 between fibroblasts from $BC004004^{-/-}$ mice and that of the wild-type controls. We also assessed the phosphorylation of Smad3, one of the major molecules which mediate the intracellular signaling of TGF-$\beta$ activation. As shown in Fig 3G, TGF-$\beta$1 induced phosphorylation of Smad3 in fibroblasts from $BC004004^{-/-}$ mice as well as cells from $BC004004^{+/+}$ mice by Western blotting analysis. However, the level of phosphorylated Smad3 induced by TGF-$\beta$1 in cells deficient in BRAP homologous protein was less than that of wild-type cells. Meanwhile, the increase in Smad3 phosphorylation induced by TGF-$\beta$1 was abolished after being treated with SIS3, a selective inhibitor of TGF-$\beta$1–dependent Smad3 phosphorylation and Smad3-mediated signaling (Fig 3H). We also examined the effect of TGF-$\beta$1 treatment on BRAP expression in wild-type cells. BRAP expression was increased after TGF-$\beta$1 stimulation for 24 h (Fig 3I). Given the results of those lung tissue examination and in vitro cell experiments, the attenuated pulmonary fibrotic lesion by bleomycin treatment in $BC004004^{-/-}$ mice might be caused by at least two mechanisms, one is the reduced TGF-$\beta$1 level in lungs and the other is the decreased viability of fibroblasts and their decreased responsiveness to TGF-$\beta$1 stimulation.

In our previous study (Liu et al, 2016), we found that down-regulation of BRAP expression by siRNA gene silencing enhanced nuclear factor-kappa B (NF-$\kappa$B) transcriptional activity in cultured human bronchial cell line 16HBE14o-. Therefore, we sought to examine whether BRAP homologous protein in mouse lung fibroblasts has the regulatory effect on NF-$\kappa$B pathway because NF-$\kappa$B is also among the key regulators of fibroblast functions. The result was shown in Fig S3. Western blotting using two different antibodies against total p65 (RelA) did show much difference between

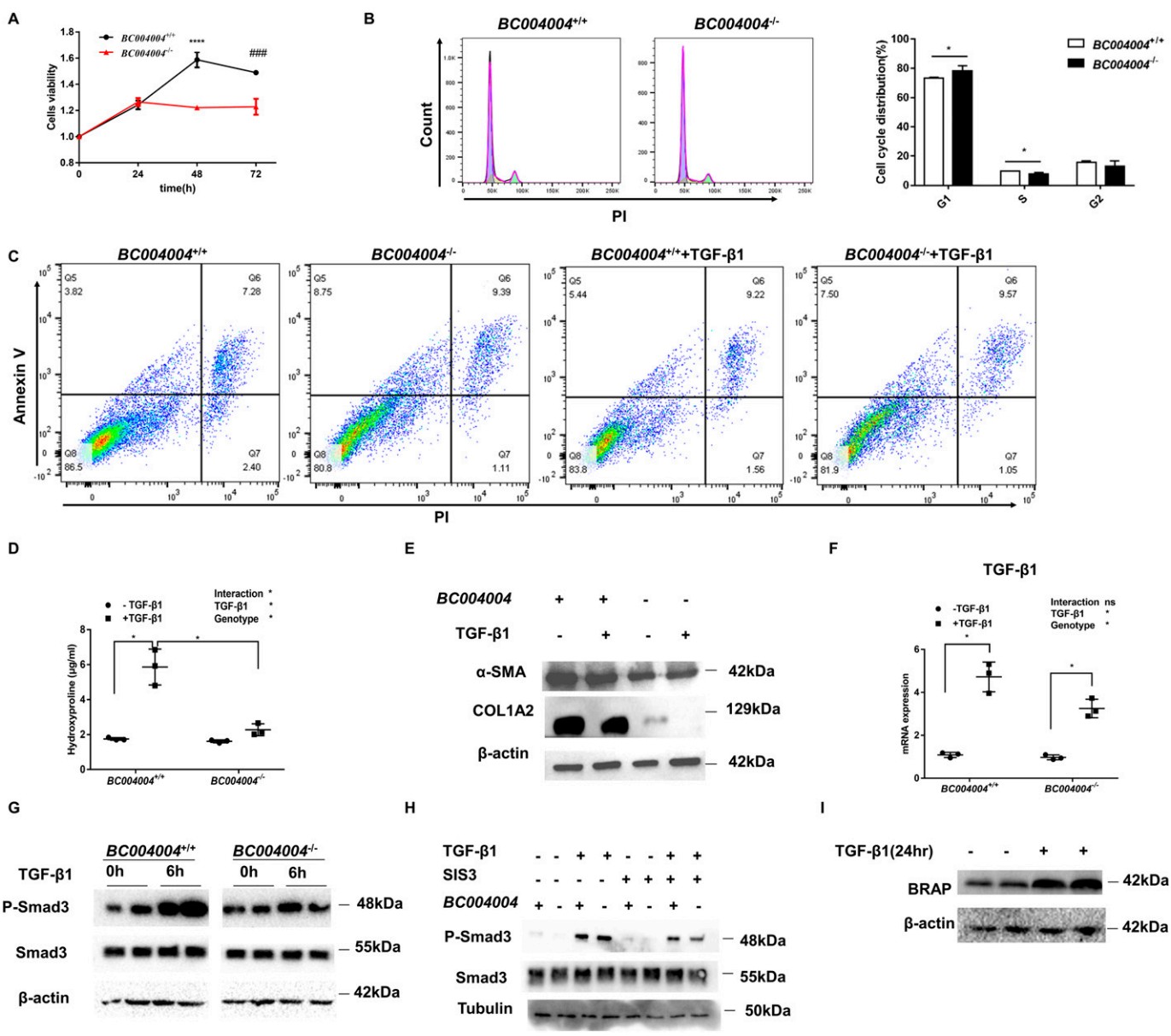

**Figure 3. Bombesin receptor–activated protein homologous protein deficiency led to decrease of collagen production in isolated fibroblasts from mouse lungs.**
**(A)** Fibroblasts were isolated from mouse lungs and cultured in vitro. Stimulation of TGF-β1 induced cell proliferation was assayed by CCK8 method. 48 or 72 h after TGF-β1 of 10 ng/ml induction cells from *BC004004*[−/−] mice showed decreased cell proliferation compared with *BC004004*[+/+] mice. n = 6. ****$P < 0.0001$, ###$P = 0.001$. **(B)** Cell cycle of isolated lung fibroblasts analyzed by flow cytometry. n = 4. *$P < 0.05$. **(C)** Apoptosis of isolated lung fibroblasts detected by FITC Annexin V and PI staining and analyzed by flow cytometry. n = 3. **(D)** The hydroxyproline content of isolated fibroblasts were measured after cells were treated with TGF-β1. n = 3, *$P < 0.05$. **(E)** Representative data of three independent experiments for Western blotting analyzing the lysates of isolated fibroblasts with antibody against α-smooth muscle actin and COL1A2. **(F)** qRT-PCR analysis of mRNA expression after TGF-β1 stimulation for 24 h in fibroblasts isolated from mouse lungs. n = 3. **(G)** Representative data of three independent experiments for Western blotting analyzing the lysates of fibroblasts from mouse lungs after TGF-β1 treatment for 6 h. **(H)** Representative data of three independent experiments for Western blotting analyzing the lysates of fibroblasts from mouse lungs in the presence of SIS3. Primary fibroblasts were treated with vehicle (DMSO) or 10 μM SIS3 for 30 min followed by the addition of 10 ng/ml TGF-β for 6 h. **(I)** Representative images of three independent experiments of Western blotting analysis of lung fibroblasts from wild-type mice. Fibroblasts were induced by TGF-β1 (10 ng/ml) for 24 h.
Source data are available for this figure.

*BC004004*[−/−] cells and wild-type control. There was no much difference on p65 phosphorylated at S536, either. Because the amount of IκBα decreased in *BC004004*[−/−] cells compared with wild-type control, we extracted the nuclear protein of fibroblasts and examined the amount of p65 translocated into the nucleus. There was

no much difference on the translocated p65 between *BC004004*[−/−] cells and control cells. However, we cannot rule out the regulatory role of BRAP homologous protein on NF-κB pathway in those isolated lung fibroblast because the amount of RelB, a member of NF-κB transcription factor family, decreased significantly in

BC004004[−/−] cells. Given that the physiological existence and relevance for the possible dimeric complexes formed by Rel proteins have not yet been fully elucidated, it is quite challenging to clarifying the role of BRAP homologous protein on the function of lung fibroblasts in BLM induced lung injury and fibrosis via NF-κB pathway.

## ATG5 is one of the physical interacting partners of BRAP

To further probe the biological roles of BRAP and investigate the underlying mechanisms, we performed an exhaustive yeast two-hybrid screening to screen potential candidates that interact with human BRAP protein. A bait plasmid pGBKT7-BRAP-CTD was constructed by inserting a truncation of *C6orf89* gene open reading frame into plasmid pGBKT7. This plasmid encoded a BRAP fragment lacking the predicted transmembrane region within the N terminus fused with the GAL4 DNA binding domain and a c-Myc epitope tag in yeast. It was transformed into Y2HGold cells which were then mated with Y187 cells containing normalized universal human cDNA library. The mated culture was spread on agar plates containing SD/−Leu/−Trp media and incubated at 30°C for 3 d to allow diploid colonies containing both bait and library plasmids to grow. Then those colonies on the plates were further replica-plated onto agar plates containing SD/−Ade/−His/−Leu/−Trp media and X-α-Gal. The blue colonies that grew on those selective plates were patched out on fresh plates containing the same media and X-α-Gal and were further picked and frozen for further sequencing analysis. We got about 6,000 blue colonies that may contain the potential binding partners for CTD fragment of BRAP protein. 4,500 colonies were already sequenced and the sequencing analysis of the remaining blue colonies is still underway when this manuscript was submitted for publishing. Among the sequenced colonies 356 genes that were potential interacting partners of BRAP were identified and the confirmation analysis of them is also underway. The information of those screened genes was provided as a supplementary table to this article. Among the dataset of potentially interacting genes, *ATG5* was further analyzed because there are five colonies containing different fragments of ATG5 cDNA. Those screened fragments encoded 1–142 aa, 1–158 aa, 1–167 aa, 1–161 aa, and 1–165 aa of ATG5 protein, respectively (Supplementary table https://doi.org/10.5281/zenodo.6601020).

To verify the interaction between the whole length of ATG5 and the CTD of BRAP, we constructed the plasmid pGADT7-ATG5 for the expression of ATG5 protein fused with a GAL4AD domain. This plasmid was transformed into Y187 cells and then mated with Y2HGold cells transformed with pGBKT7-BRAP-CTD. The formed diploids plated on agar plates containing SD/−Ade/−His/−Leu/−Trp media and X-α-Gal were growing and also turn blue, indicating the physical interaction exists between the whole ATG5 and BRAP (Fig 4A).

To confirm whether BRAP protein interacts with ATG5 in mammalian cells, we carried out CoIP assays using cultured human bronchial epithelial cell line 16HBE14o- because we found BRAP is abundant in this cell line in the previous study. As shown in Fig 4B, by using anti-ATG5 antibody to immunoprecipitate proteins that are bound to ATG5 we detected BRAP signal by Western blotting (upper panel). An anti-BRAP antibody could also pull down ATG5 in the

CoIP assay as shown in the lower panel of Fig 4B. To test whether ATG5 makes direct contact with BRAP, we expressed the GST-tagged recombinant ATG5 protein and the MBP-tagged BRAP-CTD in the *Escherichia coli* Rosetta (DE3) cells, respectively, and then performed a pull down assay. In this assay, the recombinant GST-tagged ATG5 was bound to glutathione agarose resin and the beads were thoroughly washed. The beads were then incubated with *E. coli* cell lysates containing the expressed recombinant MBP-tagged C-terminal domain of BRAP. After thoroughly washed the proteins bound to the beads were fractionated on SDS–PAGE gel and analyzed by Western blotting using the antibody against BRAP. As shown in Fig 4C, the antibody against BRAP detects a signal in the immunoblot, indicating that ATG5 binds BRAP directly.

## Lacking BRAP homologous protein led to enhanced autophagy activity

ATG5 is an essential factor for autophagy. It is constantly conjugated to ATG12 (Mizushima et al, 1998) and is involved in autophagic vesicle formation. We put forward a hypothesis that the mechanisms underlying the regulatory effect of BRAP on BLM induced lung injury might also be via the autophagy pathway by interacting with ATG5. In isolated lung fibroblasts from BC004004[−/−] mice there was also a modest decrease in ATG5 protein (Fig 4D). We examined the mRNA expression of ATG5 of those cells to explore whether the decreased level of ATG5 protein in BC004004[−/−] mice is due to down-regulation of transcription of *ATG5* gene by qRT-PCR analysis (Fig 4E). There was no significant difference for the mRNA expression of ATG5 between fibroblasts from BC004004[−/−] mice and that of the wild-type controls. Next, we sought to find a possible link between BRAP homologous protein and autophagy because BRAP was shown to interact with ATG5 in cultured human cells. LC3 and p62, the markers for autophagy activity, were assessed by immunofluorescence staining on the lung tissue sections of BC004004[−/−] mice and the wild-type control mice. The LC3 of the lung tissue sections was stained by an anti-LC3 antibody and a CY5-conjugated secondary antibody (Fig 5A). Fluorescence area was measured and calculated as the percentage of total tissue area (%) was shown in the right panel of Fig 5A. There was an increase in the amount of LC3 signal in the bronchial epithelia and the interstitial cells of BC004004[−/−] mice compared with the wild-type mice. The increase in LC3 expression in lung tissues of BC004004[−/−] mice was more robust than that of wild-type control mice after bleomycin treatment as shown in Fig 5A. p62 binds LC3 and the level of p62 serves as a selective substrate of autophagy. We performed the immunofluorescence staining of p62 on the tissue sections from mice before and after bleomycin instillation (Fig 5B). Fluorescence area was measured and calculated as the percentage of total tissue area (%) was shown in the right panel of Fig 5B. After bleomycin instillation, the decreased level of p62 in lungs of BC004004[−/−] mice compared with that of wild-type mice were not statistically significant.

The increased amount of total LC3 on lungs of BC004004[−/−] mice indicates a possible link between BRAP homologous protein with autophagic activity. Therefore, we tried to clarify whether there is enhanced autophagic activity in lung fibroblasts from BC004004[−/−] mice. We detected LC3 in isolated fibroblasts from mouse lungs by

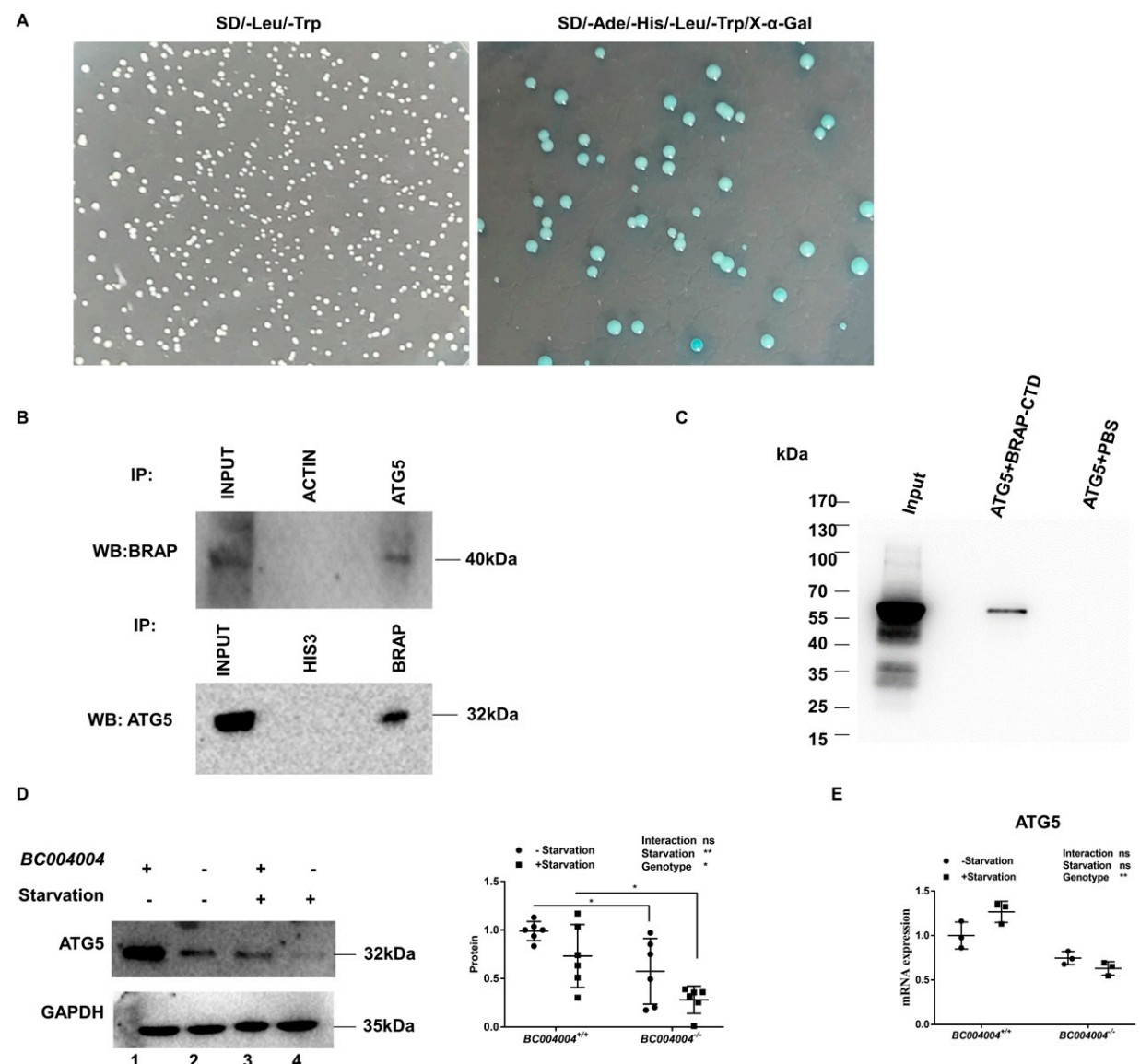

**Figure 4. Bombesin receptor–activated protein (BRAP) interacts with ATG5 in cultured human bronchial epithelial cell line 16HBE14o.**
**(A)** The interaction between full length ATG5 and the CTD of BRAP analyzed by yeast two-hybrid assay. The diploids containing both full length ATG5 and the CTD of BRAP formed white colonies on agar plates containing SD/−Leu/−Trp media. Those diploids growing on agar plates containing SD/−Ade/−His/−Leu/−Trp media turned blue with the presence of X-α-Gal, indicating the physical interaction exists between the full length ATG5 and BRAP. **(B)** Co-immunoprecipitation analysis of lysates of cultured 16HBE14o- cells using either anti-ATG5 antibody or anti-BRAP antibody to immunoprecipitate cell lysates. 5% of input of cell lysates, control experiments using an anti-actin antibody or an anti-histone antibody, and the immunoprecipitated samples were loaded in every other well on a 10% polyacrylamide gel, and then Western blotting analysis using either the anti-BRAP antibody (upper panel) or the anti-ATG5 antibody (lower panel) were performed to detect signals. **(C)** Interaction of recombinant GST-tagged ATG5 and MBP-tagged CTD of BRAP by in vitro pull down assay. Glutathione agarose resin was incubated with *Escherichia coli* cell lysates containing the recombinant GST-tagged ATG5 and then washed thoroughly. The beads was then incubated with *E. coli* cell lysates containing the MBP-tagged CTD of BRAP and washed to pull down CTD of BRAP. The samples were loaded in every other well on 10% polyacrylamide gel. 10 μl of cell lysates containing recombinant MBP-tagged CTD of BRAP was used as the input sample, which is shown in lane 1. The pull down sample is shown in lane 2. And the control experiment using glutathione agarose resin bound with the GST-tagged ATG5 to incubate with equal volume of PBS instead of cell lysates containing MBP-tagged CTD of BRAP is shown in lane 3. The immunoblot was probed with antibody against BRAP. The molecular weight of recombinant MBP-tagged CTD of BRAP is about 74 kD (the CTD of BRAP is around 32 kD). **(D)** Representative images of Western blotting analysis of lung fibroblasts isolated from mice. After starvation for 24 h, cells from individual mouse were lysed and then loaded in the wells next to the ones without starvation on 10% polyacrylamide gel (left panel). The right panel shows the quantification of ATG5 protein bands from six independent experiments before and after starvation. n = 6 in each group, *P < 0.05. **(E)** qRT-PCR analysis of mRNA expression of ATG5 in lung fibroblasts from mice. n = 3. Source data are available for this figure.

immunoblotting. Endogenous LC3 was detected as two bands after SDS–PAGE fractionation and immunoblotting. The bands migrate around 16 kD usually represent LC3-I which is cytosolic and will

convert into the active autophagosome membrane-bound LC3-II during the autophagic process. The amount of LC3-II which migrates at ~14 kD is closely correlated with the number of autophagosomes

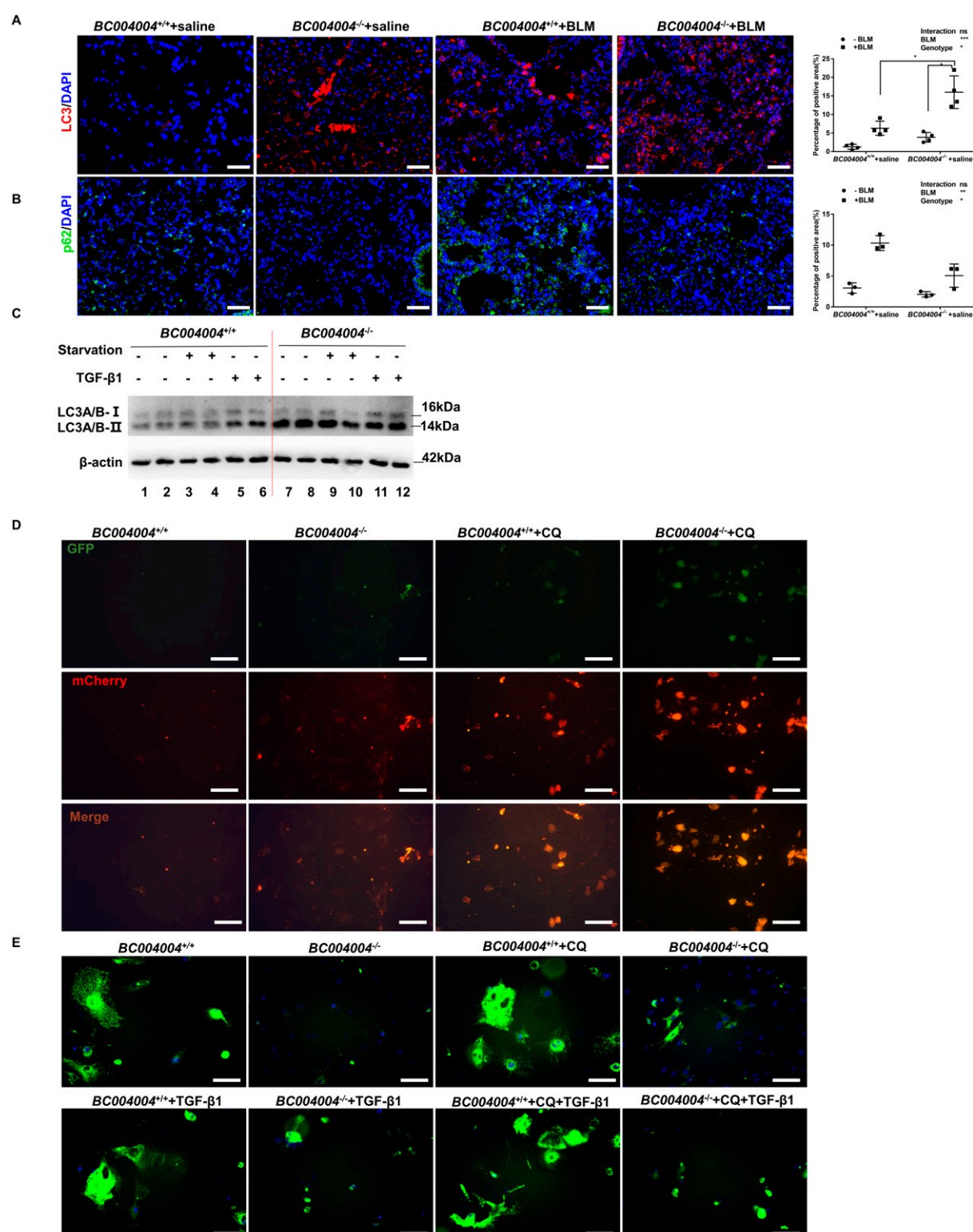

**Figure 5. Lack of bombesin receptor–activated protein homologous protein led to enhanced autophagy activity.**
**(A)** Immunofluorescence assay of mouse lung tissue sections on Day 21 after bleomycin instillation using anti-LC3 antibody and a CY5-conjugated secondary antibody (original magnification ×200). Scale bar = 100 μm. Fluorescence area of immunofluorescence assay with the anti-LC3 antibody was measured and calculated as the percentage of total tissue area (%) (one tissue section from one animal). n = 4. *P < 0.05. **(B)** Immunofluorescence assay of mouse lung tissue sections on Day 21 after bleomycin instillation using anti-p62 primary antibody and an FITC fluorescent secondary antibody (original magnification ×200). Scale bars = 100 μm. Fluorescence area of immunofluorescence assay with the anti-p62 antibody was measured and calculated as the percentage of total tissue area (%) (one tissue section from one

and thus serves as an indicator of autophagosome formation. The amounts of LC3-II in fibroblasts from lungs of $BC004004^{-/-}$ mice were more abundant under different conditions (starvation or TGF-$\beta$1 stimulation) than those of wild-type control mice, indicating more autophagosomes formed in cells without BRAP homologous protein expression (Fig 5C).

During the dynamic process of autophagy, LC3 is incorporated into the autophagosome membrane and then degraded after lysosomal fusion. This process can be monitored by the autophagic flux assay with the analysis of the fluorescence of a fusion protein mCherry-GFP-LC3B or a GFP-p62 fusion protein. As shown in Fig 5D, primary lung fibroblasts were transduced with adenovirus expressing LC3B fused with GFP and mCherry (Ad-mCherry-GFP-LC3B). This fusion protein shows signals of GFP and mCherry before the fusion with lysosomes. The expressed GFP of the fusion protein loses fluorescence because of lysosomal acidic conditions, whereas the mCherry of the fusion protein does not after fusion with lysosomes. Weak GFP and mCherry signals were found in the fibroblasts from $BC004004^{+/+}$ because of low autophagic activity, whereas the signal of mCherry increased in the fibroblasts from $BC004004^{-/-}$ mice. After the cells were treated with a lysosomal inhibitor chloroquine, the yellow puncta (merged by GFP and mCherry fluorescence) were observed in the cells from both $BC004004^{+/+}$ and $BC004004^{-/-}$, suggesting the formation of early autophagosomes and the blocking of the fusion of autophagosomes with lysosomes. BRAP deficiency increased the signal of the yellow puncta after the lysosomal inhibition by chloroquine, indicating an increased autophagic flux in the cells from $BC004004^{-/-}$. Moreover, primary lung fibroblasts were transduced with adenovirus expressing GFP-p62 fusion protein (Ad-GFP-p62) as shown in Fig 5E. There were more signals of GFP-p62 in the cells from $BC004004^{+/+}$ mice, indicating lower autophagic activity since less p62 was degraded after the fusion of autophagosomes with lysosomes. Compared with the control mice, BRAP deficiency attenuated the accumulation of the green puncta of GFP-p62 with or without the treatment with chloroquine, indicating increased autophagy in the absence of BRAP. In the cells from $BC004004^{-/-}$ mice TGF-$\beta$1 stimulation led to increased GFP signals, suggesting an inhibition effect of TGF-$\beta$1 on autophagy.

We also sought to detect the intracellular organelles of fibroblasts by transmission electron microscopy. As shown in Fig 6A, lung fibroblasts from $BC004004^{-/-}$ mice contained more autophagosomes and lysosomes compared with cells from wild-type control mice. Furthermore, we measured the activity of cathepsin B in isolated lung fibroblasts. Cathepsins are usually characterized as members of the lysosomal cysteine proteases activated at the low

pH in lysosomes. They are responsible for driving proteolytic degradation within the lysosome and have an integral role in autophagy and other lysosome dependent biogenesis. The cathepsin B activity was enhanced in lung fibroblasts from $BC004004^{-/-}$ mice compared with wild-type control (Fig 6B), indicating enhanced activity of lysosomal cysteine proteases.

Taken together, those findings demonstrated mice lacking BRAP homologous protein exhibited enhanced autophagy activity in lungs, especially in fibroblasts of lungs.

### Enhanced autophagic activity may contribute to less amount of collagen in isolated lung fibroblasts

Having established that lacking BRAP homologous protein attenuated bleomycin-induced pulmonary fibrosis and enhanced autophagic activity in lung fibroblasts, we sought to determine whether enhanced autophagic activity contributes to less collagen content in fibroblasts. As shown in Fig 6C, LC3-II increased after starvation in lung fibroblasts from wild-type mice, indicating enhanced autophagic activity induced by starvation. And the level of COL1A2 decreased after starvation in those cells. After TGF-$\beta$1 treatment the amount of COL1A2 was also less in fibroblasts from $BC004004^{-/-}$ mice compared with $BC004004^{+/+}$ mice (Fig 6D). As shown in Fig 6E, the increase in COL3A1 expression induced by TGF-$\beta$1 was abolished by Smad3 phosphorylation inhibitor SIS3. Rapamycin promotes the autophagy activity by inhibiting mTOR pathway and the treatment of rapamycin could inhibit the COL3A1 expression induced by TGF-$\beta$1 (Fig 6E).

These results suggest that enhancement of autophagic activity may contribute to less collagen production in isolated fibroblasts from mouse lungs. And lacking BRAP homologous protein led to enhanced autophagy activity. Therefore, the resistance of $BC004004^{-/-}$ mice to bleomycin-induced pulmonary fibrotic injury might be at least partially due to the enhanced autophagy caused by BRAP homologous protein deficiency. Further investigation is needed to provide more insight into the link between autophagy activity and the collagen produced in fibroblasts.

## Discussion

The antibody we used in IHC analysis for detection of BRAP on the human lung tissue samples was a recombinant monoclonal antibody. It was generated according to the fragment of 100–250 aa encoded by human $C6orf89$ gene. We found that this antibody can

---

animal). n = 3. **(C)** Representative images of three independent experiments of Western blotting analyzing lung fibroblasts from mice with an antibody against LC3. The upper bands migrating around 16 kD represent LC3-I and the lower bands around 14 kD represent LC3-II. **(D)** Representative images of three independent experiments with the analysis of the fluorescence of a fusion protein mCherry-GFP-LC3B in primary fibroblasts. Cells were transduced with Ad-mCherry-GFP-LC3B adenovirus for 24 h and then incubated with fresh medium without fetal bovine serum for another 24 h. Then the cells were treated with 25 $\mu$M of chloroquine (CQ) for 24 h before visualization of the expression of mCherry and GFP of LC3B fusion protein with fluorescence microscopy. Scale bar = 100 $\mu$m. **(E)** Representative images of three independent experiments with the analysis of the fluorescence of a GFP p62 fusion protein in primary fibroblasts. Cells were transduced with Ad-GFP-p62 adenovirus for 24 h and then incubated with fresh media without fetal bovine serum for another 24 h. Then the cells were treated with 25 $\mu$M of chloroquine (CQ) for 24 h before visualization of the expression of GFP of p62 fusion protein with fluorescence microscopy (the upper panel). Or the cells were treated with 25 $\mu$M chloroquine (CQ) for 24 h followed by TGF-$\beta$1 stimulation for 24 h before visualization of the expression of the expression of GFP of p62 fusion protein with fluorescence microscopy (the lower panel). Scale bar = 50 $\mu$m.
Source data are available for this figure.

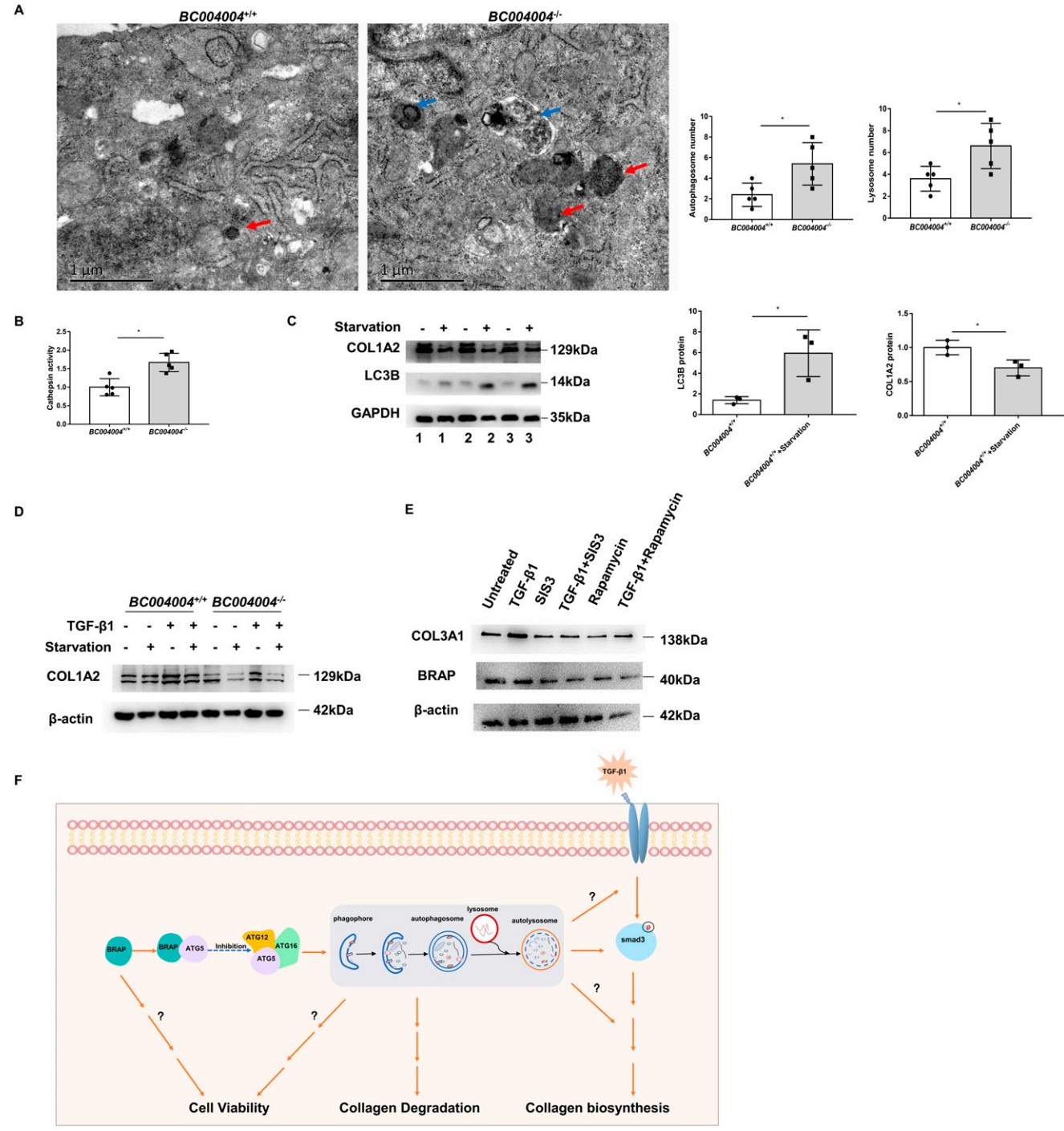

**Figure 6. Starvation-induced enhancement of autophagic activity may contribute to less collagen production in isolated fibroblasts from mouse lungs.**
**(A)** Representative images of transmission electron microscopy of isolated fibroblasts from mouse lungs. Autophagosomes were indicated by blue arrows and lysosomes were indicated by red arrows. Scale bars = 1 $\mu m$. n = 5. *$P$ < 0.05. **(B)** The cathepsin B activity in isolated lung fibroblasts from $BC004004^{-/-}$ mice and the wild-type control. n = 5*. $P$ < 0.05. **(C)** Representative images of Western blotting analysis of lung fibroblasts from three individual wild-type mice numbered 1–3. Cells from each mouse were induced by starvation for 24 h, and then were loaded in the wells next to the ones without starvation on 10% polyacrylamide gel. Relative intensity of individual bands was measured and compared with the intensity of GAPDH loading control (%) using the Bio-Rad Image Lab Analyzer software. (n = 3. *$P$ < 0.05). **(D)** Representative images of three independent experiments of Western blotting analysis of lung fibroblasts with antibody against COL1A2. Cells were treated by starvation (in the fresh media without fetal bovine serum) in the presence of TGF-$\beta$1 (10 ng/ml) or not for 24 h. **(E)** Representative images of three independent experiments of Western blotting analysis of lung fibroblasts with antibodies against bombesin receptor–activated protein or COL3A1. Primary fibroblasts were either treated with vehicle (DMSO), 10 $\mu$M of SIS3, or 1 $\mu$M of rapamycin for 30 min followed by the addition of 10 ng/ml TGF-$\beta$ for 6 h. **(F)** A summary diagram illustrating the possible signaling cascades regarding the function of bombesin receptor–activated protein in the regulation of collagen production in the fibroblasts.
Source data are available for this figure.

be used in Western blotting analysis of protein extracts from both human and mouse tissues, and in IHC analysis of human tissue samples only. Therefore, we could detect BRAP homologous protein in extracts from mouse lung tissues or isolated mouse cells by Western blotting, but could not detect BRAP homologous protein on mouse tissue sections by immunohistochemical staining. Using this antibody, we also found BRAP signals in neurons of the brain tissue samples. In addition to BLM-induced lung injury model, we found that chronic unpredictable mild stress evoked more significant depressive-like effect in $BC004004^{-/-}$ mice and caused morphological changes in the neurons of hippocampus in those mice. But the underlying mechanisms are not explored yet (data not published). The current study for BLM-induced lung fibrosis was the first study trying to explore the mechanism underlying the biological role of BRAP in cells.

Although data from experiments using BRAP null mice showed a reduced pulmonary fibrosis phenotype induced by bleomycin, and there was BRAP signal in many interstitial cells in human fibrotic lung tissues we do not know whether BRAP expression was up-regulated in fibrotic lung disease or not because we could not get healthy human lung tissue samples. As far as we know, the research work regarding the role of BRAP has been carried out in animal models and in cultured human cell lines. Basically, there are no data to clearly demonstrate that BRAP-related signaling cascades revealed in this study also occur in human diseases.

We sought to further elucidate the role of this protein in pulmonary fibrotic process. First, the reduced pulmonary inflammation in $BC004004^{-/-}$ mice might contribute to attenuated fibrotic phenotype. A decreased level of TGF-$\beta$1 in lung tissues in $BC004004^{-/-}$ mice could be one of the results of the reduced inflammation caused by bleomycin, whereas it could also contribute to attenuated fibrosis because less TGF-$\beta$1 were available to promote fibrotic process. The reason why mice lacking BRAP homologous protein manifested reduced inflammation needs to be further investigated because this protein may exist in many cell types. So a phenotype of null mice could be an effect of regulation of complex network involving many cells.

Second, the proliferation rate of the isolated fibroblasts from $BC004004^{-/-}$ mice decreased and the production of collagen in fibroblasts from $BC004004^{-/-}$ mice after TGF-$\beta$1 stimulation decreased substantially, indicating that the impaired function of fibroblasts contributes at least partially to attenuated pulmonary fibrosis phenotype of $BC004004^{-/-}$ mice. In this study, we only chose fibroblasts among those cells expressing BRAP, to further analyze the mechanism underlying the fibrotic process related to BRAP function. We explored several possible cellular signaling cascades or events involving BRAP. The first possible reason we considered is the altered NF-$\kappa$B pathway because in our previous study, we found that BRAP overexpression could inhibit NF-$\kappa$B transcriptional activity in cultured 16HBE14o- cells. However, the amounts of p65 translocated into the nucleus did not show much difference between $BC004004^{-/-}$ cells and control cells although the amount of I$\kappa$B$\alpha$ decreased in $BC004004^{-/-}$ cell. In addition, the amount of RelB, a member of NF-$\kappa$B transcription factor family, decreased significantly in $BC004004^{-/-}$ cells. BRAP homologous protein seems to have a regulatory role on NF-$\kappa$B pathway in lung fibroblasts. But it needs further investigation to elucidate to what degree that this

altered pathway is involved in the attenuated pulmonary fibrosis phenotype in $BC004004^{-/-}$ mice.

The second possible reason for the attenuated collagen production by fibroblasts could be the altered signaling cascade triggered by TGF-$\beta$1. The level of phosphorylated Smad3 induced by TGF-$\beta$1 in fibroblasts without BRAP homologous protein is less than that of wild-type cells. Yet another possible mechanism for impaired production of collagen in those fibroblasts could be enhanced autophagic activity in the absence of BRAP homologous protein. ATG5 was revealed by a modified yeast two-hybrid screening as a partner of BRAP that interacts with it directly. This modified yeast two-hybrid assay was performed to screen as many as possible the potential interacting partners for BRAP. The idea of this modification is to allow all the colonies to grow out first on plates with select agar medium only for diploids containing both bait plasmid and library plasmid. Then those colonies were replica plated for selection of interaction markers. By doing this diploid colonies could grow as many as possible as long as the diploids contained both bait and library plasmids no matter there were interactions between bait and library plasmids or not. After those diploids containing both bait and library plasmids grew and formed colonies we then selected those colonies for interactions between bait and library plasmids by plating them on selective media for interactions. The idea of this method is to avoid of stringent growing conditions for colonies on selective media of interaction markers at the very beginning. A similar idea was applied to the conduction of transformation of plasmids conferring resistance to G418 into yeast cells. After the transformation of such plasmids, the yeast cells need to be spread out on YPD plates with G418 to allow the colonies to grow out. If the transformed yeast cells were spread out on plates with atrophic selection medium combined with G418 after transformation there could be no colonies coming out because the growth condition will be too stringent for the transformed yeast cells to grow into colonies. In addition to ATG5, the yeast two-hybrid screening assay performed by our laboratory revealed more than 300 potential interacting genes with human BRAP, which were still under investigation. The major reason that we chose ATG5 for further study in this work is that we found ATG5 has a quite strong interaction with BRAP. First, we found five different cDNA fragments of ATG5 in yeast two-hybrid screening. Next, in vitro pull down assay showed that ATG5 has direct contact with BRAP. Then, compared with other CoIP experiments, we used less cell extracts to get a signal showing the interaction between ATG5 and BRAP (one could also argue that the antibodies we used are pretty good for this purpose). Finally, the expression level of ATG5 changed significantly in $BC004004^{-/-}$ mice. ATG5 is constantly conjugated to ATG12, an ubiquitin-like protein, and the ATG12-ATG5 conjugate exhibits E3 ligase like activity (Otomo et al, 2013). The conjugated ATG12-ATG5 interacts with ATG16 to form the ATG12-ATG5-ATG16 complex which localizes to nascent autophagosomes and functions upstream to the LC3 lipidation and its autophagosome targeting (Levine & Kroemer, 2008; Deretic & Levine, 2009; Lystad et al, 2019). In this study, we found that there was a decrease in p62 level and an increase in LC3 on lung tissue sections in $BC004004^{-/-}$ mice, indicating the autophagic activity was enhanced in lungs deficient in BRAP homologous protein expression. However, we cannot differentiate the cell type in which enhanced autophagy occurs

because BRAP expresses not only in bronchial epithelial cells but also in interstitial cells, alveolar macrophages, and some of the alveolar epithelial cells (data not shown). Among different cell types in lung tissue, we only tested isolated fibroblasts. TEM analysis showed more autophagosomes and lysosomes in lung fibroblasts from $BC004004^{-/-}$ mice and PE conjugated LC3-II was up-regulated in those cells, indicating enhanced autophagic activity occurs in fibroblasts. Autophagy has a number of vital roles in cells and is dynamically regulated. Autophagic activity is usually low under basal conditions, but can be up-regulated by a variety of different stimuli (Rubinsztein, 2006; Levine & Kroemer, 2008; Mizushima et al, 2008, 2010; Deretic & Levine, 2009; Lystad et al, 2019). One of the consequences of enhanced autophagic activity in fibroblasts could be reduced level of collagen in kidneys as reported by some studies (Li et al, 2016; Hsu et al, 2019; Kaushal et al, 2020). TGF-$\beta$ was found to promote autophagy in human atrial myofibroblastsis, which was required for fibrogenesis (Ghavami et al, 2015). However, there was evidence of decreased autophagic activity in lung tissues from idiopathic pulmonary fibrosis patients. TGF-$\beta$ was shown to inhibit autophagy in a human lung fibroblast cell line (Patel et al, 2012). We found that upon nutrient deprivation, the collagen content of lung fibroblasts decreased accompanied by up-regulated LC3-II level, which indicates enhanced autophagy. Starvation-induced autophagy also decreased COL1A2 production induced by TGF-$\beta$1. However, whether enhanced autophagy contributes to decreased TGF-$\beta$ level in lung tissues in $BC004004^{-/-}$ mice was not assessed in this study. The exact molecular mechanism for decreased level of TGF-$\beta$1 in $BC004004^{-/-}$ mice still needs to be further investigated.

BRAP was identified in a bacteria two-hybrid search as a potential partner for bombesin receptor subtype-3 (BRS-3) in the previous study, an orphan receptor belonging to mammalian bombesin receptor family (Liu et al, 2011). We did not examine the expression of BRS-3 in BLM induced lung injury because the interaction between BRAP and BRS-3 could not be further demonstrated by co-immunoprecipitation experiment in another study of our group (regarding the role of BRAP in the brain because BRAP was expressed in neurons of hippocampus, unpublished yet).

Taken together, our attempts to investigate the underlying mechanisms showed at least several possible pathways involved that may affect lung fibroblasts functions by BRAP deficiency. Several scenarios would occur because of the presence of BRAP in fibroblast cells as illustrated in a summary diagram in Fig 6F. BRAP inhibits autophagosomes formation by interacting with ATG5. The increased autophagy activity due to BRAP deficiency may have several effects. One of them is to perturb the signaling cascades triggered by TGF-$\beta$, and thus decrease the biosynthesis of collagen. Another effect is to degrade the collagen made by the cell. And increased autophagy may also impair the viability of fibroblasts. BRAP may also promote cell viability through mechanisms other than inhibiting autophagy.

One question that is not addressed in this study is why ATG5 is down-regulated in isolated fibroblasts from $BC004004^{-/-}$ mice. The increased level of ATG5 was reported to be related to collagen COL5A1 mRNA expression in airways of patients with refractory asthma (Poon et al, 2017). The decreased level of ATG5 could be a response to the deficiency of BRAP, which might compensate for the disturbed autophagic activity in those cells. Although we could not rule out the other explanations for attenuated lung fibrosis in the absence of BRAP because hundreds of potential interacting partners with BRAP were revealed by the yeast two-hybrid screen, the findings of this study demonstrated that the physical interaction between BRAP and ATG5 is of particular interest in regulating cellular events and thus suggest a potential novel intracellular mechanism for regulating autophagy by BRAP.

# Materials and Methods

### Animals

All animal experiments were performed in accordance with protocols approved by the Institutional Animal Care and Research Advisory Committee of Central South University. All mice used in this study were on C57BL/6J background. $BC004004^{-/-}$ mice were generated by CRISPR/Cas9–mediated genome engineering method on C57BL/6 background by a project entrusted to Nanjing Biomedical Research Institute of Nanjing University (project number: XM 002961). The schematic strategy of genome engineering was shown in Fig S1. The animals were genotyped by PCR, followed by sequence analysis to confirm the successful knockout of $BC004004$. $BC004004^{-/-}$ mice and its isogenic wild-type control C57BL/6 mice ($BC004004^{+/+}$), weighing between 17 and 20 g (male, 6 wk old), were used in this study. Mice were housed in micro-isolation cages on a constant 12-h light/12-h dark cycle in a temperature (22~23°C) and humidity (50~60%) controlled room and with free access to water and food.

### Mouse model of bleomycin-induced lung injury

Bleomycin sulphate (Bristol Laboratories) was dissolved in sterile 0.9% saline. All animals received intratracheal instillations of either bleomycin or equivalent volume of saline on day 0 as previously described (Kremer et al, 1999; Laxer et al, 1999; Berkman et al, 2001). Briefly, mice were randomly assigned to the following four groups (1) $BC004004^{+/+}$ receiving intratracheal instillations of saline. (2) $BC004004^{+/+}$ receiving intratracheal instillations of bleomycin, indicated by "$BC004004^{+/+}$+BLM" in the figures. (3) $BC004004^{-/-}$ receiving intratracheal instillations of saline. (4) $BC004004^{-/-}$ receiving intratracheal instillations of bleomycin ("$BC004004^{-/-}$+BLM" in the figures). To facilitate BLM administration animals were anesthetized with an intraperitoneal (IP) injection of 1% pentobarbital sodium (100 mg/kg), and then treated with a single intratracheal instillation of 50 $\mu$l of saline containing 3.5 mg/kg BLM. Control animals received intratracheal instillation of 50 $\mu$l of sterile saline alone. Body weights were monitored throughout the whole experiment. After intratracheal bleomycin or saline instillation, animals were euthanized by an overdose of pentobarbital (100 mg/kg, IP). Lungs were excised and subsequently subjected for histological examination or extraction of total protein or RNA. Lung injury was evaluated by analysis of histopathological examination of excised lung tissue and measurement of hydroxyproline concentration in the lung.

## Histopathology and immunohistochemistry analysis

After fixation in 4% paraformaldehyde, the lung tissues were embedded in paraffin (Liu et al, 2016) and tissue sections of 3 μm thick were prepared for further analysis. Hematoxylin-eosin (H&E), Masson's trichrome, and Sirius Red staining (Rittie, 2017) were conducted by Wuhan Saville Biotechnology. The severity of lung fibrosis was evaluated by the Ashcroft scale analysis (Wang et al, 2021). Immunohistochemistry analysis was performed according to the method described previously (Liu et al, 2016). Tissue sections were deparaffinized, hydrated, and subjected to antigenic retrieval by incubation with Tris–EDTA Buffer (10 mM TrisBase, 1 mM EDTA Solution, and 0.05% Tween 20, pH 9.0) at 98°C for 20 min. Sections were treated with 3% $H_2O_2$ in deionized water for 10 min to block endogenous peroxidase activity and then incubated with primary antibodies overnight at 4°C. Subsequently, the slices were thoroughly washed with TBS buffer containing 0.025% Triton X-100 and then incubated with HRP-conjugated goat anti-rabbit IgG (Cat. no.: PV-6002; ZSGB-Bio) for 30 min at room temperature. After three washes with PBS containing 0.025% Triton X-100, the sections were incubated with DAB reagent (Cat. no.: AR1027-1; Boster) for 2 min at room temperature. Sections were counterstained with hematoxylin (Cat. no.: G1004; Servicebio) and mounted in GVA (Cat. no.: ZLI-9551; ZSGB-Bio).

For immunofluorescent analysis with BRAP antibody, the sections were visualized using Alexor Fluor 488 AffiniPure Donkey Anti-Rabbit fluorescent secondary antibody at a dilution of 1:200 (Cat. no.: AB_2313584; Jackson ImmunoResearch), and nuclei were counterstained with DAPI in immunofluorescence at a dilution of 1:10 (Cat. no.: G1012; Servicebio). A double immunofluorescence procedure was carried out to examine the co-distribution of BRAP and S100A4 in the HLF cells grown on the coverslips. The coverslips were washed gently with PBS and fixed in freshly prepared 4% paraformaldehyde–neutral PBS at room temperature for 15 min. The coverslips were washed to remove the fixative and then incubated in 0.1% Triton X-100 in PBS at room temperature for 15 min to permeablize the cells. After being washed in PBS the coverslips were then blocked in 5% BSA for 30 min at 37°C and then incubated with primary antibodies against BRAP and S100A4 (both antibodies at a dilution of 1:200) in the blocking buffer in a humidified chamber overnight at 4°C. The coverslips were then washed three times in PBS-T (5 min for each wash) and then incubated with both secondary antibodies (Alexor Fluor 488 AffiniPure Donkey Anti-Rabbit antibody and Alexor Fluor 594 AffiniPure Goat Anti-Mouse secondary antibody) at a dilution of 1:200 for 1 h at room temperature in the dark. Then the coverslips were washed three times with PBS for 5 min each in the dark and the cell nuclei were counterstained with DAPI. After the coverslips were mounted with a drop of mounting media, the cells were examined under a fluorescence microscope.

## Hydroxyproline assay of lung tissues

To estimate the total amount of collagen in the lung, hydroxyproline was measured by Hydroxyproline assay kit (Cat. no.: A030-2-1; Nanjing Jiancheng Bioengineering Institute) according to the manual of the kit. Briefly, 30 μg of lung tissue was homogenized in 0.3 ml of cold PBS using a hand-held homogenizer. 50 μl of

**Table 1. Primers used in this study.**

| Primer | Up | Down |
|---|---|---|
| β-ACTIN | TTGCAGCTCCTTCGTTGCC | GACCCATTCCCACCATCACA |
| ATG5 | GAAGAGGAGCCAGGTGATGA | GTGGTTCCATCTAGCGAGGA |
| TGF-β1 | TCAGACATTCGGGAAGCAGT | TCGAAAGCCCTGTATTCCGT |

digestion buffer was added to a 250 μl of the homogenate and the mixture was incubated at 37°C for 3 h. Then 500 μl of Solution 1 was added to the mixture and incubated at room temperature for 10 min, followed by addition of 500 μl of Solution 2 and incubated at RT for 5 min. Then 1 ml of solution 3 was added to the mixture and incubated at 60°C for 15 min. Then the mixture was centrifuged at 1,200g for 5 min (Heraeus Fresco 17; Thermo Fisher Scientific). The sample was centrifuged at 1,200g for 5 min and the supernatant was taken for testing. Absorbance of the supernatant was then measured at 550 nm and the amount of hydroxyproline was determined according to the manual.

## Gene expression analysis by quantitative real-time reverse-transcription polymerase chain reaction (qRT-PCR)

Total RNA was extracted from lung tissues or isolated cells using RNAiso Plus (Cat. no.: 9108; Takara) according to the manufacturer's instructions. 1.0 μg of total RNA was reversely transcribed into cDNA using the Prime Script RT reagent Kit with gDNA Eraser (Cat. no. RR047B; Takara). Quantitative real-time PCR analyses were conducted using SYBR Premix Ex Taq II system (Cat. no.: RR420A; Takara) on a deep well Real-Time PCR Detection System (CFX96 Touch; Bio-Rad). Each sample was analyzed in triplicate. Primer sequences used in this study are shown in Table 1. The relative expression of mRNA was determined by normalizing the expression of each gene to ACTB gene by the $2^{-\Delta\Delta}$CT according to the previous study (Livak & Schmittgen, 2001). The primer sequences used in this study are shown in Table 1.

## Western blotting

Lung tissue samples were lysed by RIPA lysis buffer (Cat. no.: P0013B; Beyotime) containing 1% PMSF (Cat. no.: P7626; Sigma-Aldrich). Cultured cells were harvested and lysed in 1× SDS Loading Buffer (Cat. no.: 0091; DingGuo). The lysates were centrifuged at 13,800g for 10 min at 4°C and the concentration of protein in the supernatant was determined using BCA Protein Assay Kit (Cat. no.: P0006; Beyotime). After heating at 95°C for 10 min, the proteins were fractionated by electrophoresis on 10% polyacrylamide gels and transferred to polyvinylidene fluoride membrane. Then Western blotting were performed according to the method previously described (Qu et al, 2013).The following antibodies and their corresponding dilutions were used in this study: horseradish peroxidase–labeled goat anti-rabbit antibody (Cat. no.: 4050-05, 1: 5,000; SouthernBiotech), anti–β-actin mouse monoclonal, antibody (Cat. no.: A5441, 1:10,000; Sigma-Aldrich); BRAP (Cat. no.: ab181073, 1:1,000; Abcam); Smad2/3 (Cat. no.: 8685, 1:1,000; Cell Signaling Technology); P-Smad3 (Cat. no.: 52903, 1:2,000; Abcam), α-SMA (Cat. no.: ab124964, 1:2,000; Abcam); ATG5 (Cat. no.: 133158, 1:2,000; Santa

Cruz Biotechnology); TGF-$\beta$1(Cat. no.: ab64715, 1:2,000; Abcam), LC3(Cat. no.: 12741, 1:1,000; Cell Signaling Technology), p62(Cat. no.: 5,114, 1:1,000; Cell Signaling Technology), COL1A1(Cat. no.: sc-293182, 1:1,000; Santa Cruz Biotechnology), COL1A2 (Cat. no.: D120244, 1:1,000; Sangon), COL3A1(Cat. no.: ab184993, 1:2,000; Abcam); GAPDH (Cat. no.: 60004-1, 1:10,000; Proteintech); HIS3 (Cat. no.: GB11102; Servicebio); NF-$\kappa$B p65 (Cat. no.: 48676, 1:500; Signalway Antibody); RelB (Cat. no: ab154957, 1:1,000; Abcam); NF-$\kappa$B p65 (Cat. no.: ab31481, 1:1,000; Abcam); NF-$\kappa$B p65 (phospho S536) (Cat. no.: ab86299, 1:1,000; Abcam); LaminB1 (Cat. no.: ab133741, 1:1,000; Abcam); S100A4 (Cat. no.: GB11397, 1:1,000; Servicebio).

## Isolation and culture of fibroblasts from mouse lung tissue

Lung fibroblasts were isolated from mice of 1-wk-old according to the method described previously (Edelman & Redente, 2018). Lungs were minced to small pieces and digested in 0.25% trypsin–EDTA solution containing 1 mg/ml of collagenase for 40 min at 37°C. The digested tissue was filtered through a 70 $\mu$m BD falcon cell strainer and the filtered solution was centrifuged at 60$g$ for 10 min. The cell pellet was resuspended in DMEM/F12 media supplemented with 15% fetal bovine serum, seeded in six-well tissue culture plate at $5 \times 10^4$ cells/cm$^2$ and incubated in a humidified $CO_2$ incubator with 5% of $CO_2$ at 37°C. Experiments were performed with cells of passages 5–9. Cells were synchronized with media containing 0.5% fetal bovine serum and antibiotics for 24 h before the treatment with 10 ng/ml of recombinant mouse TGF-$\beta$1 (Cat. no.: 7666-MB-005/CF; R&D Systems).

## CCK8 assay

The cell proliferation of isolated and cultured fibroblasts was measured using Cell Counting Kit 8 (CCK8) assay (Cat. no.: C0038; Beyotime). 5,000–10,000 cells were plated in one well of a 96-well culture microplate with a clear bottom. Test compounds were added into cells and incubate for 1 h in a 37°C and 5% $CO_2$ incubator according to the manufacturer's instructions. The absorbance was measured at 450 nm using a microplate reader (TECAN).

## Autophagic flux assay

10,000 isolated fibroblast cells were seeded in one well of 96 well plate and cultured in DMEM/F12 media supplemented with 15% fetal bovine serum. For the measurement of mCherry-GFP-LC3B fluorescence, the cells of one well were transduced with 1.25 MOI of Ad-mCherry-GFP-LC3B (Adenovirus expressing mCherry-GFP-LC3B fusion protein; Beyotime, Cat. no.: C3011) for 24 h. For the measurement of fluorescence of GFP p62 fusion protein, the cells of one well were transduced with 1.25 MOI of Ad GFP p62 (adenovirus expressing GFP p62 fusion protein; Beyotime, Cat. no.: C3015) for 24 h, instead. The culture medium was then discarded and supplied with fresh DMEM/F12 medium without fetal bovine serum and incubated for another 24 h. Then the cells were treated with 25 $\mu$M chloroquine for 24 h before visualization of the expression of mCherry and GFP of LC3B fusion protein or the expression of GFP of p62 fusion protein with fluorescence microscopy.

## Yeast two-hybrid assay

The truncation of human *C6orf89* gene open reading frame was cloned into the bait plasmid pGBKT7 in frame with the GAL4 DNA binding domain in an attempt to identify the proteins that may interact with a BRAP fragment lacking the predicted transmembrane region within the N terminus. The design of the truncation is based on the mRNA of the NCBI Reference Sequence NM_152734 and the truncation encodes the C-terminal most 271 aa (84–354 amino acids) subsequently. The predicted transmembrane region spans from 66 to 86 amino acids from the N terminus according to the sequence of NM_152734. The resulted construct pGBKT7-BRAP-CTD contains the above truncation within Nde I and BamH I sites. Then the construct was transformed into Y2HGold cells and the expression of BRAP CTD fragment containing 84–354 aa of BRAP was confirmed by Western blotting using both antibodies against BRAP and c-Myc epitope tag. The yeast two-hybrid assay was performed using Matchmaker Gold Yeast Two-Hybrid System (PT4084-1; Clontech, Cat. no.: 630489) according to the manufacturer's instruction with the following modifications. After mating of Y2HGold cells transformed with pGBKT7-BRAP-CTD (bait culture) and Y187 cells containing normalized universal human Mate & Plate Library (Cat. no.: 630480; Clontech) according to the manual, the mated culture was spread on 200 of 100 mm agar plates containing SD/−Leu/−Trp media and the plates were incubated at 30°C for 3 d. Then those plates were replica-plated onto 200 of 100 mm agar plates containing SD/−Ade/−His/−Leu/−Trp and 40 $\mu$g/ml of X-$\alpha$-Gal and incubated at 30°C for 3–5 d. All the blue colonies that grew on the plates were patched out onto fresh plates containing the same media and X-$\alpha$-Gal using flat sterile toothpicks. The total number of blue colonies that contain the potential binding partners for CTD fragment of BRAP protein is about 6,000 and they were frozen at −80°C for further analyses. The plasmids from the isolated colonies were rescued using Yeast Plasmid Mini Preparation Kit (Cat. no.: D0029; Beyotime) and the prey inserts were sequenced using T7 primer to retrieve the information of corresponding genes.

Because colonies containing *ATG5* gene identified in the screen only encode peptides fragments of ATG5 protein, to verify the interaction between the full length of ATG5 and BRAP by yeast two-hybrid, we constructed the plasmid pGADT7-ATG5 for the expression of ATG5 fused with a GAL4AD domain in yeast cells. This plasmid was transformed into Y187 cells and then mated with Y2HGold cells which were transformed with a bait plasmid pGBKT7-BRAP-CTD. The formed diploids were then plated onto agar plates containing SD/−Ade/−His/−Leu/−Trp medium and X-$\alpha$-Gal. If there is interaction between ATG5 and the C-terminal fragment of BRAP, the colonies will grow on the plates and will also turn blue.

## Co-immunoprecipitation assay

16HBE14o- cells (Cozens et al, 1994), kindly provided by Dr. Dieter C Gruenert, University of California, San Francisco, were cultured and used in co-immunoprecipitation experiments. Cells were cultured at 37°C with 5% $CO_2$ in Dulbecco's modified Eagle's medium containing 10% fetal bovine serum, 100 U/ml penicillin, and 100 mg/ml of streptomycin. $1 \times 10^7$ of 16HBE14o- cells were harvested with 0.25% trypsin–EDTA, rinsed twice with cold PBS, then lysed in 500 $\mu$l

of cold RIPA buffer (50 mM Tris–HCl, pH 7.6, 150 mM NaCl, 1% Nonidet P-40, 0.5% sodium deoxycholate, and 0.1% SDS) containing 1 mmol of PMSF, 1 mmol of DTT, and 1 mmol of protease inhibitor on a shaker for 30 min at 4°C. Magnetic Dynabeads Protein A beads (Cat. no.: 10001D; Invitrogen) were equilibrated three times with RIPA buffer before incubation with supernatant and antibody. After centrifuge at 12,000$g$ for 20 min at 4°C, 200 $\mu$l of supernatant were incubated with 1 $\mu$g of antibody and 25 $\mu$l of equilibrated beads by rotating at 4°C overnight. Then beads were washed three times with 1 ml of ice-cold NP-40 buffer (150 mM NaCl, 1% Igepal [NP-40], and 50 mM Tris·HCl [pH 8.0]). 50 $\mu$l of 1× SDS loading Buffer was added to the beads, boiled for 5 min and further analyzed by immunoblotting.

### Extraction of nuclear protein from isolated lung fibroblasts

The nuclear protein of isolated lung fibroblasts was extracted with NE-PER Nuclear and Cytoplasmic Extraction Reagents (Cat. no.: 78833; Thermo Fisher Scientific) as the protocol provided by the manufacturer. Briefly, $1 \times 10^7$ of cells were treated with TGF-$\beta$1 for 30 min and then harvested with trypsin–EDTA digestion and then centrifuge at 500$g$ for 5 min. The cells were washed three times with cold PBS. The cells were fully resuspended in 200 $\mu$l of CER I solution by vortexing the tube vigorously on the highest setting for 15 s and then were incubated on ice for 10 min. Then 11 $\mu$l of the ice-cold CER II solution was added to the tube, The tube was incubated on ice for 1 min and then vortexed for 5 s on the highest setting. The tube was centrifuged at 16,000$g$ for 5 min and the supernatant (cytoplasmic extract) was removed to a clean tube.100 $\mu$l of NER solution was added to the pellet and the tube was vortexed on the highest setting for 15 s. Then the sample was placed on ice and continued to be vortexed for 15 s every 10 min, for a total of 40 min. The sample was centrifuged at 16,000$g$ for 10 min and the supernatant (nuclear extract) was transferred to a prechilled tube and stored at –80°C until use.

### Flow cytometric analysis of isolated lung fibroblasts

Cell cycle analysis of fibroblasts was performed as previously described with the following modifications (Qu et al, 2013). Briefly, the isolated lung fibroblasts of $BC004004^{-/-}$ mice or from $BC004004^{+/+}$ mice were harvested and washed with PBS, and then fixed with pre-cooled 70% ethanol at 4°C overnight. On the next day, the cells were washed with ice-cold PBS, digested with 50 mg/ml DNase-free RNAase, and stained with 50 mg/ml propidium iodide (PI) solution (Cat. no.: C1052; Beyotime) at 37°C for 30 min. The populations of G1, S, and G2 cells were then quantified by flow cytometry.

Apoptosis of isolated lung fibroblasts was analyzed using FITC Annexin V Apoptosis Detection Kit I (Cat. no.: 556547; BD Pharmingen) as described by the manufacturer. The cells were resuspended in binding buffer and stained with Annexin V-FITC, PI, and both in combination. The cells were analyzed by flow cytometry within 1 h. Percentage of cells identified as apoptosis is determined by gating for double positivity for Annexin V-FITC and PI staining.

### Recombinant proteins expression and the pull down assay

To express GST-tagged recombinant ATG5 from *E. coli*, the ATG5 coding sequence was cloned into the pGEX4T1 vector to construct plasmid pGEX4T1-ATG5. The human BRAP CTD coding sequence from plasmid pGBKT7-BRAP-CTD was subcloned into a pMAL vector (New England Biolabs) to give pMAL-BRAP-CTD, which can express MBP-tagged BRAP-CTD in *E. coli*. The above plasmids were transformed into the *E. coli* Rosetta (DE3) cells, respectively. A 200 ml culture of cells containing each plasmid was grown to an $A_{600}$ of 0.5, followed by induction with 1 mM isopropylthiogalactoside (IPTG) for 2 h at 37°C before the cells were harvested by centrifuging at 900$g$ for 10 min. The cell pellet was stored at –20°C. On the next day, the cells were resuspended in 10 ml of ice-cold PBS containing 1% of Triton X-100 and 1 mM of PMSF and then sonicated 5 s × 90 times with 15 s pauses at 130W. The lysates were kept on ice at all times. The lysate containing GST-tagged recombinant ATG5 was allowed to bind 100 $\mu$l of Glutathione agarose resin (50% slurry, Cat. no.: 16100; Pierce) for 1 h at 4°C with gentle agitation. Then the slurry was centrifuged at 500$g$ for 10 min. The supernatant was discarded. The beads were washed 10 min each for three times in 1 ml of ice-cold PBS, followed by incubating with lysate containing MBP-tagged BRAP-CTD for 1 h at 4°C with gentle agitation. After centrifugation at 500$g$ for 10 min, the beads were washed 10 min each for three times in 1 ml of ice-cold PBS. The bound products were resolved in 50 $\mu$l of 2× SDS sample buffer and fractionated by SDS–PAGE gel electrophoresis. Immunoblotting with BRAP rabbit monoclonal antibody was then performed to detect BRAP CTD fragment.

### Cathepsin B activity assay

Cathepsins are members of the lysosomal cysteine protease family which plays an important role in proteolysis. Cathepsin B is one of the important lysosomal cysteine proteases that may enhance the activity of other proteases and thus may play an essential role for degradation of ECM components. We measured the activity of cathepsin B activity using Magic Red Cathepsin assay kit (Cat. no.: ICT937; Bio-Rad) according to the manufacturer's protocol. Briefly, primary lung fibroblasts were isolated from either $BC004004^{+/+}$ mice or $BC004004^{-/-}$ mice and cultured at the density of $2 \times 10^6$ cells/ml. Then the cells were incubated with the staining solution containing a cathepsin B target sequence peptide (RR)2 linked to a red (Cresyl Violet) fluorescent probe for 30 min at 37°C protected from light. Red fluorescence was generated as Cathepsin B protease activity progresses. Magic Red substrate has an optimal excitation and emission wavelength tandem of 592 and 628 nm, respectively. After incubation, the fluorescence intensity of the red fluorescent generated within the cells was analyzed on Varioskan system fluorescent plate reader (Thermo Fisher Scientific).

### Statistical analyses

Statistical analyses were performed on GraphPad Prism 7. The results are expressed as the mean of independent experiments ± SD. Two-way ANOVA was used to analyze the effects of two factors (e.g., BLM treatment or genotype). If there was interaction between those factors, further test were then conducted for the main effect

of each factor and Tukey test was applied for pairwise comparisons between groups. If no significant interaction was found between the two factors, and the main effect of one factor is significant, the Tukey test was then applied to determine whether the changes is related to an individual factor. In the latter case, the effect of one factor on corresponding changes is not dependent on the other factor. A $P$-value of 0.05 or less was considered statistically significant.

## Supplementary Information

## Acknowledgements

This work was primarily supported by Natural Science Foundation of Hunan Province Grant 2020JJ4688 and National Natural Science Foundation of China (NSFC) Grant 81570026 (to X Qu). It was also supported by the following grants: NSFC Grant 81970033 (to X Qin), Natural Science Foundation of Hunan Province Grant 2020JJ4776 (to Y Xiang), NSFC Grant 82070034 and Open Foundation of Hunan College Innovation Program Grant 20K142 (to C Liu), NSFC Grant 31900424 and Natural Science Foundation of Hunan Province Grant 2019JJ50760 (to L Pan), and the Open Sharing Fund for the Lager-scale Instruments and Equipment of Central South University.

### Author Contributions

H Wang: conceptualization, resources, data curation, software, formal analysis, validation, investigation, visualization, methodology, and writing—original draft.
W Zhang: investigation.
R Liu: investigation.
J Zheng: data curation, formal analysis, and investigation.
X Yao: data curation.
H Chen: software and investigation.
J Wang: data curation and investigation.
HC Weber: conceptualization and writing—review and editing.
X Qin: conceptualization and funding acquisition.
Y Xiang: software and funding acquisition.
C Liu: software and funding acquisition.
H Liu: software and project administration.
L Pan: software and funding acquisition.
X Qu: conceptualization, data curation, formal analysis, supervision, funding acquisition, investigation, methodology, project administration, and writing—original draft, review, and editing.

### Conflict of Interest Statement

The authors declare that they have no conflict of interest.

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
