## [Reviewer comments · Life Science Alliance]

Life Science Alliance

Lack of bombesin receptor-activated protein attenuates bleomycin induced pulmonary fibrosis in mice

Hui Wang, Wenrui Zhang, Rujiao Liu, Jiaoyun Zheng, Xueping Yao, Hui Chen, Jie Wang, Horst Weber, Xiaoqun Qin, Yang Xiang, Chi Liu, Huijun Liu, Lang Pan, and Xiangping Qu

DOI: <https://doi.org/10.26508/lsa.202201368>

Corresponding author(s): *Xiangping Qu, Central South University*

Review Timeline:

Submission Date:	2022-01-10
Editorial Decision:	2022-02-06
Revision Received:	2022-06-02
Editorial Decision:	2022-06-24
Revision Received:	2022-06-27
Accepted:	2022-06-28

Scientific Editor: Novella Guidi

Transaction Report:

February 6, 2022

Re: Life Science Alliance manuscript #LSA-2022-01368

Dr. Xiangping Qu
Central South University
Department of Physiology
Changsha, -- None selected -- 41000
China

Dear Dr. Qu,

Thank you for submitting your manuscript entitled "Lack of bombesin receptor-activated protein attenuates bleomycin induced pulmonary fibrosis in mice" to Life Science Alliance. The manuscript was assessed by expert reviewers, whose comments are appended to this letter. We, thus, encourage you to submit a revised version of the manuscript back to LSA that responds to all of the reviewers' points.

Thank you for this interesting contribution to Life Science Alliance. We are looking forward to receiving your revised manuscript.

Sincerely,

B. MANUSCRIPT ORGANIZATION AND FORMATTING:

Reviewer #1 (Comments to the Authors (Required)):

This study found that deficiency of BRAP mouse homogenous protein attenuates bleomycin-induced lung fibrosis. The knock out mice have produced less TGF-beta and their lung fibroblasts do not response well to TGF-beta treatment, both of which may contribute to their protection from bleomycin-caused lung fibrosis. In addition, the BRAP homolog was found directly binding ATG5, which is believed to down regulate autophagic activities. The authors suggest that in the absent of BRAP homolog, autophagy activities are increased to protect bleomycin-treated lungs. The study is highly novel, and the data are reasonably solid. However, there are a few pieces of the puzzle are still missing.

1. The authors provided evidence that BRAP homolog affects both TGF-beta signaling and autophagic activities. However, there are no data to illustrate if TGF-beta signaling and autophagic activities interact with each other. Since the BRAP homolog directly binds to ATG5, a plausible scenario would be that BRAP/ATG5 complex or reduced autophagic activities allow the transduction of TGF-beta signaling during fibrogenesis. On the other hand, several studies have reported TGF-beta promotes autophagic activities, which, in contradict to this study, suggest TGF-beta-promoted autophagy is required for fibrogenesis (Ghavami et al 2015). It is also possible that TGF-beta and BRAP/ATG5 do not interact with each other during lung fibrosis, because TGF-beta my interact with a BRAP complex involves one of the other 355 binding proteins to be determined by the authors.

2. The conclusion that lacking of BRAP homolog enhances autophagy process is solely based on lung immunohistological staining of total LC3 and p62, which is not sufficient. At least, autophagic flux should be determined. This is especially necessary, as BRAP homolog deficient mice have lower levels of BRAP homolog both in protein and in mRNA that WT mice, which is however, commonly associated with lower autophagic activities.

Other minor issues:

1. Histology figures and flow figures are in low resolution.

2. Bar graphs should be changed to dot plots, so that readers can better appreciate the divers of samples.

3. In several occasions, the order of Figure numbers is not consistent with it order of appearance in the text. For example, Fig 1b was cite in the text before fig 1a.

4. In many of the figures, four groups were compared, for examples: WT controls, BC004004-/- controls, WT + BLM and BC004004-/- + BLM. In these cases, one way ANOVA should be utilized with a Tukey's multiple comparisons test to determine significant differences.

5. Fig 1b, co-staining with a fibroblast marker is needed to determine if BRAP is expressed on fibroblasts.

6. Fig 7b, quantify and statistically compare the amount of autophagosomes and lysosomes.

Reviewer #2 (Comments to the Authors (Required)):

In their manuscript, Qu et al, explore a role for the bombesin receptor-activated protein (BRAP) in bleomycin (BLM) induced pulmonary fibrosis (PF) in mice. They generated a new mouse (BRAP^{-/-}), which lacks BRAP expression. They found that these mice are more resistant to PF compared with WT mice. They also isolated primary fibroblasts and found that deletion of BRAP protein affects the proliferation and collagen deposition of lung fibroblasts. Furthermore, they discovered an interaction between BRAP and ATG5 gene, which is known to be involved in autophagy process. The authors conclude that the action of BRAP on the pathogenesis of PF may be due to its involvement in autophagy process, through interaction with ATG5.

Major comments

- Statistical significance in most results should be assessed with two-way ANOVA (with post hoc Bonferroni correction) and not with t-test. The statistical test used should be included in every figure legend.
- Histology images should be accompanied by a statement mentioning the number of mice/sections that have been stained. An isotype control must be also included perhaps as supplementary info; a scale bar should be also included.
- Histology images should be accompanied by the ashcroft scoring for the evaluation of bleomycin-induced lung fibrosis and the reduction of PF upon BRAP deletion.
- All results should be presented with Dot blot graphs to clearly show the number of samples. Why there are so many fluctuations in the number of WT and knockout animals? How many times this experiment has been performed?

Minor comments

- More information about the gene and their role in diseases, especially of the lungs, should be included.
- Please justify the use of tuberculosis lung tissue. The quality of the immunostaining images should be improved and a control BRAP staining in healthy tissue should be included as control. Additionally, images with higher magnification should be included. Indicating fibroblastic areas with no appropriate staining should be avoided.
- Figure 2 is rather a supplementary one. Please add above the photos of the controls a legend (saline). b-c) Low quality images, need to be replaced. Additionally, the attenuation of PF in knockout mice compared with the controls, is not obvious in these images.
- Figure 4e: Col1a2 levels didn't increase after TGF- β 1 stimulation.
- Figure 4f: This experiment must be repeated upon TGF- β 1 stimulation to detect possible differences in stress fibers formation.
- Figure 4g: Col1a2 levels didn't increase. Please remove the graph. Additionally, col1a1 is a better marker that could be used instead.
- Figure 4i: Does not show any result, maybe supplementary.
- Figure 5d: Not convincing. Higher magnification images and controls are required.
- Figure 6a,b: The images do not show the same areas of the lung.
- Figure 7c: This experiment must be repeated upon TGF- β 1 stimulation of fibroblasts.

Reviewer #3 (Comments to the Authors (Required)):

In this study the role of bombesin receptor activated protein(BRAP) in bleomycin induced pulmonary fibrosis is investigated. Using BRAP-KO mice evidence is provided that BRAP can regulate collagen production with the result that BRAP KO mice develop less pulmonary fibrosis and that BRAP interacts with the autophagic factor ATG-5 providing evidence that BRAP is important in autophagia. Furthermore, the BRAP KO mice have less inflammation, decreased TGF-1B production, lower proliferation rates of fibroblasts, less collagen production by the fibroblasts, decreased TGF-1B stimulation of SMAD, increased autophagic activity and down regulation of ATG5.

General

This is a well-done and complete study. It would be enhanced by some editing, inclusion of support human data if it exists, as reviewed below, and also by a inclusion of a summary diagram/figure in the discussion to summarize the proposed diverse signaling mechanisms, that could be a good take away image.

Specific

1. It is unclear as written if there is any information on how BRAP activates these processes in inflammation. Is it thought to be due to an overexpression of BRAP during inflammation and is BRS-3 involved in this process? Some sentences in the introduction or discussion in this area would be helpful
2. Is there any data from human studies supporting these findings here from the BRAP KO mouse, that activation of these signaling cascades also occur in human fibrotic lung disease?
3. The discussion reiterates in detail in a number of places information from the results, hence the discussion should be carefully edited to reduce redundancy.
4. A summary diagram in the discussion summarizes the possible signaling cascades roles in the fibrosis would be a great addition because there is so much data here.

Responses to reviewers' comments:

(Reviewers' comments in bold)

Reviewer #1 (Comments to the Authors (Required)):

This study found that deficiency of BRAP mouse homogenous protein attenuates bleomycin-induced lung fibrosis. The knock out mice have produced less TGF-beta and their lung fibroblasts do not response well to TGF-beta treatment, both of which may contribute to their protection from bleomycin-caused lung fibrosis. In addition, the BRAP homolog was found directly binding ATG5, which is believed to down regulate autophagic activities. The authors suggest that in the absent of BRAP homolog, autophagy activities are increased to protect bleomycin-treated lungs. The study is highly novel, and the data are reasonably solid. However, there are a few pieces of the puzzle are still missing.

1. The authors provided evidence that BRAP homolog affects both TGF-beta signaling and autophagic activities. However, there are no data to illustrate if TGF-beta signaling and autophagic activities interact with each other. Since the BRAP homolog directly binds to ATG5, a plausible scenario would be that BRAP/ATG5 complex or reduced autophagic activities allow the transduction of TGF-beta signaling during fibrogenesis. On the other hand, several studies have reported TGF-beta promotes autophagic activities, which, in contradict to this study, suggest TGF-beta-promoted autophagy is required for fibrogenesis (Ghavami et al 2015). It is also possible that TGF-beta and BRAP/ATG5 do not interact with each other during lung fibrosis, because TGF-beta my interact with a BRAP complex involves one of the other 355 binding proteins to be determined by the authors.

Response:

We thank the reviewer for the comment. We agree that the interaction between TGF- β signaling and autophagic activities would be interesting and important. On the one hand, we tried to address the question how TGF- β signaling affects autophagic activities. However, we could not provide a clear-cut answer to this question when we did more experiments. As shown in Figure 5C in the revised manuscript, TGF- β increased the amount of LC3-II in fibroblasts from wild type mice by western blot. We did several other experiments and the results showed that TGF- β seems to either have inhibition effect on autophagy or have no effect as shown in the following Figures in this response section.

Figure 1 in the response section:

Figure 2 in the response section:

In the above Figure 1, TGF-β induced the increase of p62 in fibroblasts, which indicates that autophagic activity were inhibited by TGF-β (from results of three independent experiments). In the above Figure 2, TGF-β did not induce the increase of LC3-II amount, which indicates TGF-β did not have effect on autophagic activity.

In Figure 5E of the revised manuscript, primary lung fibroblasts were transduced with adenovirus expressing GFP-p62 fusion protein (Ad-GFP-p62). More signals of GFP-p62 in the cells from *BC004004^{+/+}* indicates lower autophagic activity since less p62 was degraded after the fusion of autophagosomes with lysosomes. In the cells from *BC004004^{-/-}* mice TGF-β stimulation led to increased GFP signals, suggesting an inhibition effect of TGF-β on autophagy. However, TGF-β stimulation did not have effect on the signals of GFP-p62 in wild type fibroblasts, indicating TGF-β signaling did not lead to inhibition of autophagy in those cells. The above results are not quite consistent regarding the effect of TGF-β on autophagy. Therefore, we could not get a clear conclusion that TGF-β signaling leads to inhibition or enhancement of autophagic activity in lung fibroblasts in this study. We think those controversial data we got might be due to the fact that autophagy is a very dynamic process with very complicated regulations.

On the other hand, we also thought about the question whether enhanced autophagic activity due to BRAP deficiency has effect on TGF-β signaling. We did experiment to show that BRAP deficiency led to reduction of Smad3 phosphorylation induced by TGF-β in isolated lung fibroblast cells (as shown in Figure 3H in the revised manuscript). And SIS3, a selective inhibitor of TGF-β1-dependent Smad3 phosphorylation and Smad3-mediated signaling, abolished the TGF-β triggered Smad3 phosphorylation, as shown in Figure 3I in the revised manuscript. However, when we tried to use inhibitors of autophagic process such as chloroquine and 3-Methyladenine, or inducers of autophagy such as rapamycin, to see whether inhibition of autophagic process or enhancement of autophagy have any

effect on the Smad3 phosphorylation induced by TGF- β 1, we could not get consistent results. Therefore, we can only draw a conclusion that BRAP deficiency led to reduction of Smad3 phosphorylation induced by TGF- β . But we don't know whether the reduction of Smad3 phosphorylation in the absence of BRAP is due to enhanced autophagy or not. We also agree with the reviewer that it is possible that TGF- β and BRAP/ATG5 do not interact with each other, because TGF- β may interact with a BRAP complex involves one of the other partners of BRAP which were revealed by yeast two-hybrid.

2. The conclusion that lacking of BRAP homolog enhances autophagy process is solely based on lung immunohistological staining of total LC3 and p62, which is not sufficient. At least, autophagic flux should be determined. This is especially necessary, as BRAP homolog deficient mice have lower levels of BRAP homolog both in protein and in mRNA that WT mice, which is however, commonly associated with lower autophagic activities.

Response:

We thank the reviewer for pointing out this. We performed autophagic flux assays with the analysis of the fluorescence of a fusion protein mCherry-GFP-LC3B as well as a GFP-p62 fusion protein using isolated primary lung fibroblasts. As shown in Figure 5D in the revised manuscript, lung fibroblasts were transduced with adenovirus expressing LC3B fused with GFP and mCherry (Ad-mCherry-GFP-LC3B). Weak GFP and mCherry signals were found in the fibroblasts from *BC004004*^{+/+} because of low autophagic activity, while the signal of mCherry increased in the fibroblasts from *BC004004*^{-/-} mice. After the cells were treated with a lysosomal inhibitor chloroquine, the yellow puncta (merged by GFP and mCherry fluorescence) were observed in the cells from both *BC004004*^{+/+} and *BC004004*^{-/-}, suggesting the formation of early autophagosomes and the blocking of the fusion of autophagosomes with lysosomes. BRAP deficiency increased the signal of the yellow puncta after the lysosomal inhibition by chloroquine, indicating an increased autophagic flux in the cells from *BC004004*^{-/-}. The lung fibroblasts were also transduced with adenovirus expressing GFP-p62 fusion protein (Ad-GFP-p62) as shown in Figure 5E in the revised manuscript. There were more signals of GFP-p62 in the cells from *BC004004*^{+/+} mice, indicating lower autophagic activity since less p62 was degraded after the fusion of autophagosomes with lysosomes. Compared with the control mice, BRAP deficiency attenuated the accumulation of the green puncta of GFP-p62 with or without the treatment with chloroquine, indicating increased autophagy in the absence of BRAP. Those results showed that BRAP deficiency increased autophagic flux in the cells from *BC004004*^{-/-}.

Other minor issues:

1. Histology figures and flow figures are in low resolution.

Response:

We thank the reviewer for pointing out this problem. We put the histology figures and flow figures in high resolution in the revised manuscript.

2. Bar graphs should be changed to dot plots, so that readers can better appreciate

the divers of samples.

Response:

We thank the reviewer for pointing out this. We replaced the bar graphs with the dot plots.

3. In several occasions, the order of Figure numbers is not consistent with it order of appearance in the text. For example, Fig 1b was cite in the text before fig 1a.

Response:

We apologize for the mistake. We carefully examined the figures of each section to make sure the numbers of the figures are consistent with the order of the revised manuscript.

4. In many of the figures, four groups were compared, for examples: WT controls, BC004004-/- controls, WT + BLM and BC004004-/- + BLM. In these cases, one way ANOVA should be utilized with a Tukey's multiple comparisons test to determine significant differences.

Response:

We thank the reviewer for pointing out this. And according to both reviewers' (reviewer #1 and reviewer #2) comments regarding the problem of the statistical analysis, we did the two-way ANOVA and provided the results (statistical significance of the p values) of the two-way ANOVA in the figures of the current manuscript, including the statistical test of the interaction between the two factors (e.g., BLM treatment and the genotype).

5. Fig 1b, co-staining with a fibroblast marker is needed to determine if BRAP is expressed on fibroblasts.

Response:

We thank the reviewer for pointing out this. The antibody against human BRAP does not work in immunohistochemistry analysis of tissue sections from mice (it works in western blot with tissue extracts or isolated cells from mice). Therefore, we used cultured human lung fibroblast cell line HLF cells to perform the co-staining. As shown in Figure 1B in the revised manuscript, there was BRAP signal in human lung fibroblast cell line HLF cells. The co-distribution of BRAP and fibroblast-specific protein 1 (FSP1, also called S100A4), which is considered a marker of fibroblasts was shown by double immunofluorescence analysis in cultured HLF cells. As shown in the lower panel of Figure 1B, green BRAP immunofluorescence overlapped with red S100A4 fluorescence and then the merged picture showed an orange signal which indicates the co-distribution of BRAP and S100A4 in the cytoplasm of fibroblasts.

6. Fig 7b, quantify and statistically compare the amount of autophagosomes and lysosomes.

Response:

We thank the reviewer for pointing out this. The amount of autophagosomes and lysosomes was quantified and statistically compared accordingly. The results were shown in Figure 6A in the revised manuscript.

Reviewer #2 (Comments to the Authors (Required)):

In their manuscript, Qu et al, explore a role for the bombesin receptor-activated protein (BRAP) in bleomycin (BLM) induced pulmonary fibrosis (PF) in mice. They generated a new mouse (BRAP^{-/-}), which lacks BRAP expression. They found that these mice are more resistant to PF compared with WT mice. They also isolated primary fibroblasts and found that deletion of BRAP protein affects the proliferation and collagen deposition of lung fibroblasts. Furthermore, they discovered an interaction between BRAP and ATG5 gene, which is known to be involved in autophagy process. The authors conclude that the action of BRAP on the pathogenesis of PF may be due to its involvement in autophagy process, through interaction with ATG5.

Major comments

- **Statistical significance in most results should be assessed with two-way ANOVA (with post hoc Bonferroni correction) and not with t-test. The statistical test used should be included in every figure legend.**

Response:

We thank the reviewer for pointing out this. And according to both reviewers' (reviewer #1 and reviewer #2) comments regarding the problem of the statistical analysis, we did the two-way ANOVA on GraphPad Prism 7 and followed the main steps of the flowchart below.

The statistical significance of the *p* values of the two-way ANOVA was indicated in the figures of the revised manuscript including the test of the interaction between the two factors (e.g., BLM treatment and the genotype).

- **Histology images should be accompanied by a statement mentioning the number of mice/sections that have been stained. An isotype control must be also included perhaps as supplementary info; a scale bar should be also included.**

Response:

We thank the reviewer for pointing out this. We usually used one tissue section to perform the IHC analysis from one mouse. We showed the numbers of the mice in the figure legends accordingly. In the revised manuscript, we used Dot blot graphs to represent all the results. Therefore, the numbers of the animals of each group we analyzed in the IHC experiments were shown by the numbers of dots in those graphs. And we also mentioned the numbers in some figure legends such as in the legends of Figure 1E and 1F. Figure 1E is the representative images for HE results of BLM treated mice and Figure 1F is the Ashcroft scoring results of BLM models for the evaluation of lung fibrosis. We did HE staining from 31 mice in each group (4 groups: BC004004^{+/+} + saline, BC004004^{-/-} + saline, BC004004^{+/+} + BLM, BC004004^{-/-} + BLM) and then determined the Ashcroft scores of each tissue section (one tissue section from one mouse). The results of the Ashcroft score were shown by Dot blot graph. For the other histology images we indicated the numbers

of animals with dots in the dot plot graph. The isotype controls for BLM induced lung fibrosis models which did not receive any treatment were provided in the supplementary figure. The scale bars were all included in the histology images.

- **Histology images should be accompanied by the ashcroft scoring for the evaluation of bleomycin-induced lung fibrosis and the reduction of PF upon BRAP deletion.**

Response:

We thank the reviewer for pointing out this. We provided the Ashcroft scores of each animal for the evaluation of lung fibrosis in Figure 1F. 31 mice were included in each group as mentioned above (4 groups: *BC004004*^{+/+} + saline, *BC004004*^{-/-} +saline, *BC004004*^{+/+} + BLM, *BC004004*^{-/-} + BLM). HE staining was performed on one tissue section from one animal and the results of the Ashcroft score were shown by dot blot graph.

- **All results should be presented with Dot blot graphs to clearly show the number of samples. Why there are so many fluctuations in the number of WT and knockout animals? How many times this experiment has been performed?**

Response:

We thank the reviewer for pointing out this. In the revised manuscript, we used dot blot graphs to represent all the results. We used 31 animals in each group to establish the BLM induce lung fibrosis. There were four groups: 4 groups: *BC004004*^{+/+} + saline, *BC004004*^{-/-} +saline, *BC004004*^{+/+} + BLM, *BC004004*^{-/-} + BLM. We did HE staining for tissue sections from all the mice used in each group (31 for each group). However, only 4-6 tissue sections randomly chosen from each group were analyzed for other histological experiments. The numbers of each experiment were indicated by the numbers of dots in those dot plot graphs.

Minor comments

- **More information about the gene and their role in diseases, especially of the lungs, should be included.**

Response:

We thank the reviewer for pointing out this. As we mentioned in the manuscript, BRAP (encoded by *C6orf89* in human and *BC004004* in mouse) was expressed in some tissues. However, the information of this gene and its related diseases is quite limited. Lalioti, V. S., et al. published a paper characterizing the gene *C6orf89* and its transcripts in 2013 (Ref 11 of the revised manuscript, J Cell Physiol 228, 1907-1921). Our group found BRAP signals in several cell types by IHC analysis of tissue sections from human samples, including bronchial epithelial cells, epithelial cells of the skin, and even in the neurons of brain tissue. BRAP antibody detected more signals in neurons compared with other cell types. However, the biological role of this protein is not clear. We established several disease models using knockout mice *BC004004*^{-/-} and got some preliminary data. We found that *BC004004*^{-/-} mice exhibited more severe phenotypes in OVA-induced asthma like disease, dextran sulfate sodium (DSS) induced colitis and chronic unpredictable mild stress (CUMS) evoked depressive-like effect and morphological changes in the neurons

of hippocampus in mice. We also found that BRAP expressed in most of the lung adenocarcinoma samples. But the underlying mechanisms are not explored yet. The current study for BLM-induced lung fibrosis was the first study trying to explore the mechanism underlying the biological role of BRAP in cells. We mentioned the potential role of this gene in the discussion section in the revised manuscript.

- **Please justify the use of tuberculosis lung tissue. The quality of the immunostaining images should be improved and a control BRAP staining in healthy tissue should be included as control. Additionally, images with higher magnification should be included. Indicating fibroblastic areas with no appropriate staining should be avoided.**

Response:

We thank the reviewer for pointing out this problem. We increased the magnification of the images. There were only few fibroblasts in the tuberculosis samples compared with that of fibrotic lung tissues. We could not get the samples of healthy tissue. The BRAP immunostaining is not that significant in the cells of tuberculosis samples compared with the fibrotic samples, so we clarified this in the revised manuscript.

According to the comment from reviewer #1, we performed a double fluorescence staining with the anti-BRAP antibody and an antibody against fibroblast-specific protein 1 (FSP1, also called S100A4) on the human lung fibroblast cell line HLF cells. S100A4 is considered as a marker for fibroblast. As shown in Figure 1B in the revised manuscript, there was BRAP signal in human lung fibroblast cell line HLF cells. The co-distribution of BRAP and S100A4 was shown in the lower panel of Figure 1B, green BRAP immunofluorescence overlapped with red S100A4 fluorescence and then the merged picture showed an orange signal which indicates the co-distribution of BRAP and S100A4 in the cytoplasm of fibroblasts.

- **Figure 2 is rather a supplementary one. Please add above the photos of the controls a legend (saline). b-c) Low quality images, need to be replaced. Additionally, the attenuation of PF in knockout mice compared with the controls, is not obvious in these images.**

Response:

We thank the reviewer for pointing out this problem. We used images with much higher resolution in the revised manuscript and add “saline” above the photos of the controls. The original histological images of Figure 2 were combined with the original Figure 3 in the revised manuscript (as shown in Figure 2 in the revised one). We did not put the original Figure 2 as a supplementary figure since we consider the images of the Masson’s staining, Sirius Red staining and IHC with COL1A2 antibody were helpful to shown the lung fibrotic changes. And we quantified the positively stained area of tissue sections from 4-6 mice and did the statistical analysis. The results were presented as dot plot graphs in the revised manuscript.

- **Figure 4e: Col1a2 levels didn’t increase after TGF- β 1 stimulation.**

Response:

We thank the reviewer for pointing out this. COL1A2 levels didn't increase after TGF- β 1 stimulation as shown in Figure 4E in the original manuscript. Sometimes the isolated primary cells do not respond well to TGF- β 1 stimulation. But just as shown in Figure 6D there was an increase of COL1A2 levels after TGF- β 1 stimulation. And as shown in Figure 6E, the COL3A1 level increased after TGF- β 1 stimulation.

•Figure 4f: This experiment must be repeated upon TGF- β 1 stimulation to detect possible differences in stress fibers formation.

Response:

We thank the reviewer for pointing out this. We did the experiments to show whether TGF- β 1 stimulation cause any effects on stress fibers formation and provided the representative images in Figure 3G in the revised manuscript. TGF- β 1 stimulation did not affect stress fibers formation.

• Figure 4g: Col1a2 levels didn't increase. Please remove the graph. Additionally, col1a1 is a better marker that could be used instead.

Response:

We thank the reviewer for pointing out this. We removed the graph in the revised manuscript. We could not get a good COL1A1 antibody in the latter experiments so we did not use it as the marker for collagen production. COL1A2 and COL3A1 were also the genes which were screened by the yeast two-hybrid to be the potential interacting partner with BRAP. There were two yeast colonies that contained fragments of COL1A2 gene in the yeast two-hybrid screening. The fragments encode 1-158aa and 1-217aa of COL1A2. And there was one colony that contained the fragment of COL3A1 gene and encodes 1-165aa of COL3A1. However, we did not find interaction between BRAP and COL1A2 or between BRAP and COL3A1 in the following yeast two-hybrid experiments using plasmids encodes full-length of the gene COL1A2 or COL3A1.

• Figure 4i: Does not show any result, maybe supplementary.

Response:

We thank the reviewer for pointing out this problem. We make a mistake for Figure 4i in the original manuscript. We forgot to put the western blot result of RelB in this figure (we described the result of RelB in the result section of the text). The RelB western blot was shown in the supplementary figure of the revised manuscript. And it is in the figure shown here. We apologize for this mistake!

- **Figure 5d: Not convincing. Higher magnification images and controls are required.**

Response:

We thank the reviewer for pointing out this problem. We quantified the positively stained area of the tissue sections (one tissue section from one mouse, 3 mice for each group) and assessed those data with two-way ANOVA. The difference between knockout mice and their wild type controls is not significant when analyzed by two-way ANOVA. Therefore, we remove this figure in the revised manuscript.

- **Figure 6a,b: The images do not show the same areas of the lung.**

Response:

We thank the reviewer for pointing out this problem. We changed representative images in the revised manuscript (Figure 5A and 5B).

- **Figure 7c: This experiment must be repeated upon TGF-β1 stimulation of fibroblasts.**

Response:

We thank the reviewer for pointing out this. We did the experiments to detect the effects of TGF-β1 stimulation on COL1A2 and COL3A1 expression in fibroblasts and provided the results in the revised manuscript (Figure 6D and 6E in the revised manuscript).

Reviewer #3 (Comments to the Authors (Required)):

In this study the role of bombesin receptor activated protein(BRAP) in bleomycin induced pulmonary fibrosis is investigated. Using BRAP-KO mice evidence is provided that BRAP can regulate collagen production with the result that BRAP KO mice develop less pulmonary fibrosis and that BRAP interacts with the autophagic factor ATG-5 providing evidence that BRAP is important in autophagia. Furthermore, the BRAP KO mice have less inflammation, decreased TGF-1B production, lower proliferation rates of fibroblasts, less collagen production by the fibroblasts, decreased TGF-1B stimulation of SMAD, increased autophagic activity

and down regulation of ATG5.

General

This is a well-done and complete study. It would be enhanced by some editing, inclusion of support human data if it exists, as reviewed below, and also by a inclusion of a summary diagram/figure in the discussion to summarize the proposed diverse signaling mechanisms, that could be a good take away image.

Specific

1. It is unclear as written if there is any information on how BRAP activates these processes in inflammation. Is it thought to be due to an overexpression of BRAP during inflammation and is BRS-3 involved in this process? Some sentences in the introduction or discussion in this area would be helpful

Response:

We thank the reviewer for pointing out this. In the revised manuscript, we provided the western blot result that TGF- β 1 stimulation for 24 hours led to increase of BRAP expression in isolated lung fibroblasts from wild type mice (Figure 3J in the revised manuscript). We did not examine the expression of BRS-3 in BLM induced lung injury. In another study of our group (regarding the role of BRAP in the brain since BRAP was expressed in neurons of hippocampus, unpublished yet) we did a co-immunoprecipitation experiment using the antibody against BRAP and an antibody against BRS-3, trying to detect the interaction between them since BRAP was first screened as a potential partner with BRS-3 by a bacteria two-hybrid assay. We could not demonstrate that there was interaction between BRS-3 and BRAP by co-IP. However, in the brain tissue sections from BRAP-KO mice IHC analysis showed a decrease of BRS-3 expression in neurons. In our previous study, overexpression of BRAP in human bronchial epithelial cell line 16HBE14o-cells led to down-regulation of NF- κ B transcriptional activity by a reporter assay and siRNA silencing of C6orf89 in those cells led to increase of NF- κ B transcriptional activity. We examined the signaling of NF- κ B in the current study but could not get a clear-cut conclusion (as shown in supplementary figure3 in the revised study). Therefore, the role of BRAP in inflammation is still not clear. We agree that just as the reviewer pointed out, it may be related to an upregulation of BRAP in inflammation. We discussed this interesting question in the discussion section.

2. Is there any data from human studies supporting these findings here from the BRAP KO mouse, that activation of these signaling cascades also occur in human fibrotic lung disease?

Response:

We thank the reviewer for pointing out this. In the fibrotic lung tissues there was BRAP signal in many interstitial cells (many of them could be fibroblasts but we did not use a specific marker for fibroblasts) as detected by the antibody against BRAP (as shown in Figure 1A). But since we could not get healthy human lung tissue samples, we don't know whether BRAP expression was up-regulated in fibrotic lung disease or not. We found that BRAP was expressed in the cancer cells of most of the lung adenocarcinoma while it only

presents in very few samples of lung squamous cell carcinoma. As far as we know, the research work regarding the role of BRAP was done in animal models and in cultured human cell lines. Basically, there is no data to clearly demonstrate that BRAP related signaling cascades revealed in this study also occur in human diseases.

3. The discussion reiterates in detail in a number of places information from the results, hence the discussion should be carefully edited to reduce redundancy.

Response:

We thank the reviewer for pointing out this. According to all the reviewers' comments and the experiments recently done, we carefully examined and edited the text of manuscript and provided the revised one.

4. A summary diagram in the discussion summarizes the possible signaling cascades roles in the fibrosis would be a great addition because there is so much data her

Response:

Thanks for the suggestion. We generated a summary diagram to summarize the possible signaling cascades related to the role BRAP in the lung fibroblast cells (as shown in Figure 6F) and discussed those signaling cascades in the discussion section of the revised manuscript.

June 24, 2022

RE: Life Science Alliance Manuscript #LSA-2022-01368R

Dr. Xiangping Qu
Central South University
Department of Physiology
110 Xiangya Road
Changsha, China

Dear Dr. Qu,

Thank you for submitting your revised manuscript entitled "Lack of bombesin receptor-activated protein attenuates bleomycin induced pulmonary fibrosis in mice". We would be happy to publish your paper in Life Science Alliance pending final revisions necessary to meet our formatting guidelines.

- address Reviewer 2's remaining points
- please use the [10 author names, et al.] format in your references (i.e. limit the author names to the first 10)
- please add a callout for Figure S1 to your main manuscript text

A. FINAL FILES:

B. MANUSCRIPT ORGANIZATION AND FORMATTING:

****It is Life Science Alliance policy that if requested, original data images must be made available to the editors. Failure to provide**

original images upon request will result in unavoidable delays in publication. Please ensure that you have access to all original data images prior to final submission.**

The license to publish form must be signed before your manuscript can be sent to production. A link to the electronic license to publish form will be sent to the corresponding author only. Please take a moment to check your funder requirements.

Sincerely,

Reviewer #1 (Comments to the Authors (Required)):

In this revised manuscript, the authors have addressed my concerns and improved the quality of the study.

Reviewer #2 (Comments to the Authors (Required)):

In the revised manuscript, Wang et al., accommodated most of our concerns.

However, a few minor issues remain to be clarified:

TGF exposure of fibroblasts is well accepted that leads to the formation of stress fibers and the upregulation of collagens expression. The authors must explain the discrepancy in their experiments, which can be due to the absence of substrate or the number of passages. Proof must be provided that their cell system (primary cells from newborn mice cultured for 5-9 passages!) behaves normally. Otherwise, some of the experiments should be removed (e.g. stress fibers).

In figure 2D the new immunostaining images of TGF- β 1 must be improved more, while they do not add much to the story and can be entirely removed. The visualization of statistical significance is confusing.

Reviewer #3 (Comments to the Authors (Required)):

The authors have addressed adequately all points I raised

Responses to Reviewers' remaining points on the Manuscript:

We gratefully acknowledge the valuable comments and constructive suggestions made by reviewers of the paper. Our responses are given in the following, some of which have been incorporated in the revised manuscript.

Reviewer #1 (Comments to the Authors (Required)):

In this revised manuscript, the authors have addressed my concerns and improved the quality of the study.

Reviewer #2 (Comments to the Authors (Required)):

In the revised manuscript, Wang et al., accommodated most of our concerns.

However, a few minor issues remain to be clarified:

TGF exposure of fibroblasts is well accepted that leads to the formation of stress fibers and the upregulation of collagens expression. The authors must explain the discrepancy in their experiments, which can be due to the absence of substrate or the number of passages. Proof must be provided that their cell system (primary cells from newborn mice cultured for 5-9 passages!) behaves normally. Otherwise, some of the experiments should be removed (e.g. stress fibers).

Our response:

We thank the reviewer for the comment. Our experiments showed that TGF- β 1 exposure of fibroblasts did not lead to the formation of stress fibers in the cells by fluorescence Phalloidin-TRITC staining of the cells. We think the reason of this phenomenon might be due to the method we handled the cells for staining. In order to stain the cells with Phalloidin-TRITC, the cells needed to be cultured on the glass coverslips. In this experiment, we did not treat the coverslips with collagens which were always used for cells to be attached well to the glass surface. The cells might not be growing on those glass coverslips quite well since the morphology of the cells was not the same with the cells on the surface of culture dishes. Those cells on the glass coverslips might be under stressed condition already and the exposure of TGF- β 1 did not lead to further formation of stress fibers. Accordingly, the figure which showed that the absence of BRAP did not lead to the formation of stress fibers in the original manuscript might also have this problem. Therefore we could not come to the conclusion that lacking BRAP protein in the cells did not affect the stress fibers formation. We thank the reviewer for pointing out this and we removed the figures related to the stress fiber formation in the revised version. We apologize for the inappropriate experiments we performed! We examined all the other experiments in this paper and there were no other experiments that used the same glass coverslips for cell culture.

In figure 2D the new immunostaining images of TGF- β 1 must be improved more, while they do not add much to the story and can be entirely removed. The visualization of statistical significance is confusing.

Our response: We thank the reviewer for the comment. We removed this figure (Figure 2D)

accordingly. The western blot using tissue extracts showed that the level of TGF- β 1 in the knockout mice *BC004004*^{-/-} after BLM treatment was much less than that of wild type mice after the BLM treatment as shown in the Figure 2F of the original manuscript (Figure 2E in the revised final version).

Reviewer #3 (Comments to the Authors (Required)):

The authors have addressed adequately all points I raised

June 28, 2022

RE: Life Science Alliance Manuscript #LSA-2022-01368RR

Dr. Xiangping Qu
Central South University
Department of Physiology
110 Xiangya Road
Changsha, China

Dear Dr. Qu,

Thank you for submitting your Research Article entitled "Lack of bombesin receptor-activated protein attenuates bleomycin induced pulmonary fibrosis in mice". It is a pleasure to let you know that your manuscript is now accepted for publication in Life Science Alliance. Congratulations on this interesting work.

DISTRIBUTION OF MATERIALS:

Again, congratulations on a very nice paper. I hope you found the review process to be constructive and are pleased with how the manuscript was handled editorially. We look forward to future exciting submissions from your lab.

Sincerely,
